# Learning to Generate Unit test via Adversarial Reinforcement Learning

**Dongjun Lee[1,†], Changho Hwang[2], Kimin Lee[1]**
dgjun32@kaist.ac.kr

KAIST[1], Microsoft Research[2]

## Abstract

Unit testing is a core practice in programming, enabling systematic evaluation of programs produced by human developers or large language models (LLMs). Given the challenges in writing comprehensive unit tests, LLMs have been employed to automate unit test generation, yet methods for training LLMs to produce high-quality unit tests remain underexplored. In this work, we propose UTRL, a novel reinforcement learning (RL) framework that trains an LLM to generate high-quality unit test given a programming instruction. Our key idea is to iteratively train two LLMs, the unit test generator and the code generator, in an adversarial manner via RL: (1) the unit test generator is trained to maximize a discrimination reward, encouraging it to produce tests that reveal faults in the code generator's solutions; and (2) the code generator is trained to maximize a code reward, encouraging it to produce solutions that pass the unit tests generated by the unit test generator. In our experiment, we demonstrate that unit tests generated by Qwen3-4B trained via UTRL show higher quality compared to unit tests generated by the same model trained via supervised fine-tuning on ground-truth unit tests, yielding code evaluations that more closely align with those induced by the ground-truth tests. Moreover, Qwen3-4B trained with UTRL outperforms frontier models like GPT-4.1 and GPT-4o in generating high-quality unit tests, highlighting the effectiveness of UTRL in training LLMs for the unit test generation. Code and relevant materials are available at our project page: https://dgjun32.github.io/UTRL.

## 1 Introduction

Unit test is a critical component in programming, as it enables evaluation and verification on the functional correctness of the program produced by human developers or large language models (LLMs). Recently, unit tests are widely used as verifiable reward functions in reinforcement learning (RL) or test-time scaling for LLM-driven code generation, where the unit tests provide task-specific feedback signals for guiding the generation of functionally correct code (Le et al., 2022; Guo et al., 2024).

However, implementing unit tests for each programming task is labor-intensive and challenging, since (1) a unit test should contain functionally valid test cases, and (2) each test case in the unit test should cover

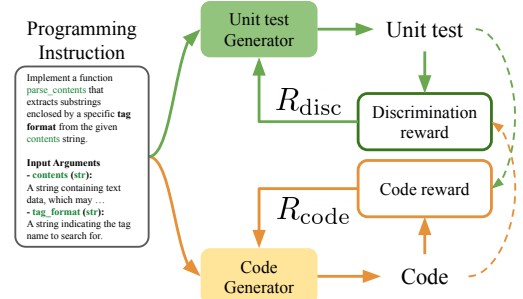

Figure 1: Overview of UTRL. The unit test generator is trained to generate unit test that detect fault in code generated by the code generator, and the code generator is trained to produce code that passes the generated unit test.

challenging edge cases, capable of discriminating subtly faulty code implementations. These require significant level of code reasoning capability and understanding of the programming task.

---

† Work done during internship at Microsoft Research.

As LLMs have shown strong code understanding and generation capabilities, recent works have explored training them to generate high-quality unit tests given a programming task. A common approach is to collect instruction–unit test pairs, followed by supervised fine-tuning (SFT). To enable scalable annotation of high-quality unit tests, these works propose strategies such as using capable teacher models (Ma et al., 2025), applying code perturbation techniques (Prasad et al., 2025), or adopting generator–validator frameworks (Wang et al., 2025b). While promising, SFT-based methods are difficult to scale across diverse programming domains as they fundamentally require unit test labels, which are costly to obtain, for every training example. In contrast, RL removes the need for explicit unit test labels by directly optimizing models against reward signals, offering a more scalable paradigm for training LLMs in unit test generation. The key challenge, however, lies in designing a reward function that can reliably assess the quality of generated unit tests without relying on ground-truth annotations.

In this work, we present **UTRL**, an adversarial RL framework iterating over training a unit test generator LLM and a code generator LLM in an adversarial manner, each LLM defining a reward signal for its counterpart. Our main contribution lies in designing the reward for training unit test generation (i.e., discrimination reward), which rewards the unit test generator LLM when the generated unit tests successfully discriminate the code solution produced by the code generator LLM from ground-truth code solution. As a result, UTRL eliminates the need for ground-truth unit test annotations, since the reward is defined by ground-truth code solution and code generated by the code generator LLM. Specifically, as illustrated in Figure 1, the unit test generator is optimized to maximize the discrimination reward, which encourages producing unit tests that can detect code generated by the code generator LLM from the ground-truth code, while the code generator is optimized to generate code that passes all test cases in the generated unit tests. Through this adversarial training, the code generator LLM progressively learns to generate code solutions that are harder to distinguish from the ground-truth code solutions, and the unit test generator LLM, in turn, learns to generate highly discriminative test cases that can detect subtle faults in the near-correct code solutions, thereby becoming capable of covering challenging edge cases.

In our experiments, we evaluate the quality of generated unit tests on the TACO evaluation set (Li et al., 2023), which contains competitive programming tasks collected from multiple online judge platforms. We first show that unit tests generated by Qwen3-4B (Yang et al., 2025) trained with UTRL, when used as a code reward function for best-of-N sampling, induce $3.1\times$ higher code accuracy gain, compared to the unit tests generated by the base Qwen3-4B. Additionally, we demonstrate that Qwen3-4B trained via UTRL achieves unit test quality higher than the same model trained via SFT, despite not relying on unit test annotations from humans or more capable models, and even surpasses frontier models such as GPT-4.1 (OpenAI, 2025). Finally, the code generator adversarially trained with the unit test generator achieves code generation performance comparable to the same model trained to maximize the pass rate over ground-truth unit tests.

## 2 RELATED WORK

**Unit test generation with LLM**   Unit testing has been extensively explored for reliable software development (Fraser & Arcuri, 2011; Alagarsamy et al., 2024). As LLMs show strong capability in programming, recent works have explored the automation of unit test generation using large language models (LLMs), with a focus on evaluating the test generation capability of LLMs (Jain et al., 2024a; Wang et al., 2024), iteratively refining the unit test by leveraging coverage analysis or mutation testing as a contextual feedback (Alshahwan et al., 2024; Altmayer Pizzorno & Berger, 2025; Dakhel et al., 2024; Foster et al., 2025), or utilizing co-evolutionary algorithm (Li et al., 2025). Another line of work investigates training LLMs for unit test generation via supervised learning, with a focus on constructing high-quality unit test annotations at scale. CodeRM (Ma et al., 2025) leverages stronger teacher models to generate unit tests automatically, while UTGEN (Prasad et al., 2025) proposes a data collection strategy that perturbs code solutions to create faulty variants, and then collects high-quality test cases that discriminate between the correct and faulty code. The work most closely related to ours is CURE (Wang et al., 2025a), which explores the co-evolution of code generation and test case generation capabilities via reinforcement learning. However, our approach differs from CURE in two key aspects: (1) Unlike CURE, which assumes access to a dataset annotated with ground-truth unit tests, UTRL requires only instruction-code pair dataset, which is readily available at large scale (Li et al., 2023; Lambert et al., 2024; Prasad et al., 2025),

and (2) UTRL is a framework aimed at training LLMs to generate an optimal set of test cases, rather than a single test case.

**Reinforcement learning for LLM**  Early works on training LLMs with reinforcement learning (RL) utilized reward models learned from human preference feedback and trained LLM with the Proximal policy optimization algorithm (PPO) (Schulman et al., 2017) for aligning LLMs with human values, establishing reinforcement learning with human feedback (RLHF) as a standard framework for instruction tuning and alignment in LLMs (Stiennon et al., 2020; Ouyang et al., 2022). To address the complexity and instability of online RL, subsequent methods proposed offline preference learning algorithms that bypass the need for reward models while maintaining strong alignment performance (Rafailov et al., 2023; Ethayarajh et al., 2024; Hong et al., 2024). More recently, RL algorithms with improved efficiency have been proposed as a variant of PPO, and such RL algorithms have been actively applied to improve the reasoning capabilities of LLMs, particularly in domains such as math and programming where verifiable rewards can be precisely defined (Shao et al., 2024; Guo et al., 2024; Yu et al., 2025). However, different from code generation or math, defining the verifiable rewards for unit test generation is non-trivial and remains underexplored.

**Adversarial learning and self-play**  Adversarial learning provides a general framework in which models improve by competing against an adaptive opponent, even enabling model to learn complex data distribution (Goodfellow et al., 2014). Building on this idea, self-play has emerged as a powerful paradigm where the opponent is derived from the model itself. A line of works has demonstrated that symmetric self-play can yield superhuman performance in strategic decision-making (Silver et al., 2017), while another line of work explores asymmetric formulations in which agents generate and solve tasks to construct an automatic curriculum (OpenAI et al., 2021). With the rapid progress of LLMs across diverse domains, self-play has recently been extended to LLM-based systems, enabling improvement with minimal or even zero additional data. AZR introduce a zero-data self-play framework in which a task generator and a task solver co-evolve to improve the model's reasoning ability (Zhao et al., 2025). In alignment, SPIN applies self-play to alignment by iteratively contrasting LLM-generated responses with human responses, enabling improvement without costly preference data (Chen et al., 2024). In code generation, Sol-Ver utilizes self-play between a solution generator and a test generator to jointly improve code and unit tests (Lin et al., 2025). In theorem proving, STP leverage self-play between a conjecture generator and a prover to enhance theorem-proving capability of LLM (Dong & Ma, 2025).

## 3 METHOD

In this section, we present UTRL, a framework for adversarially training a unit test generator LLM and a code generator LLM via RL. Section 3.1 describes an overview of the UTRL, and the core components of UTRL are described in Section 3.2 and Section 3.3.

### 3.1 OVERVIEW

Our goal is to train a unit test generator LLM $\mathcal{M}_{\text{UT}}$, which takes a programming instruction $I$ as input and outputs a unit test $\mathcal{T}$, as shown in Figure 2. A unit test $\mathcal{T}$ consists of multiple test cases, i.e., $\mathcal{T} = \{T_i\}$, where each $T_i$ denotes a single test case. Each test case $T_i$ checks the partial correctness of a code implementation $C$ for instruction $I$, by verifying that $C$ produces the expected output for a given input. A straightforward approach to training $\mathcal{M}_{\text{UT}}$ is to collect a dataset of instruction-unit test pairs and fine-tune LLMs via supervised learning. However, curating such a dataset is inherently challenging because implementing unit tests is an open-ended task with no unique solution, and it is often difficult to objectively evaluate their quality.

To address these challenges, we propose a framework for training a unit test generator LLM $\mathcal{M}_{\text{UT}}$ without relying on instruction–unit test pairs. Since instruction–code solution pairs are widely available, we train the $\mathcal{M}_{\text{UT}}$ via RL to produce unit tests that can distinguish ground-truth code from imperfect code generated by a code generator LLM, $\mathcal{M}_{\text{code}}$. We further extend this idea into an adversarial framework: $\mathcal{M}_{\text{UT}}$ improves by learning to generate unit test identifying weaknesses in code generated by $\mathcal{M}_{\text{code}}$, while $\mathcal{M}_{\text{code}}$ in turn improves by learning to produce code that passes increasingly challenging unit tests.

Specifically, we iterate over the following two key steps (see Algorithm 1):

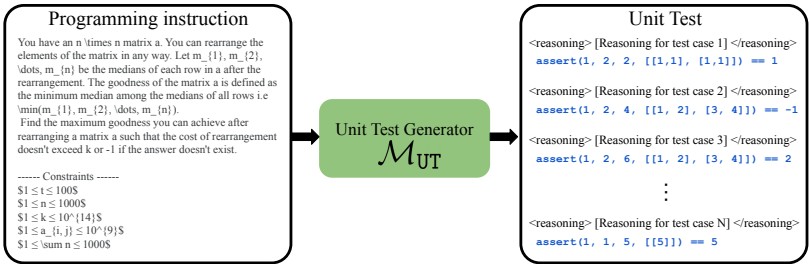

Figure 2: Illustration of input and desired output of unit test generator LLM. Given a programming instruction specifying input arguments and functionality of a code, the unit test generator LLM generates a set of $N$ test cases comprehensively covering the various edge cases based on reasoning.

---

**Algorithm 1** UTRL

---

**Require:** Dataset of instruction-code solution pairs $\mathcal{D} = \{(I_k, C_k^*)\}$,
         Initial code generator LLM $\mathcal{M}_{\text{code}}$, Initial unit test generator LLM $\mathcal{M}_{\text{UT}}$,
         Weight for the discrimination reward $\lambda$,
         Desired minimum number of test cases per unit test $\tau$
1: **while** not converged **do**
2:      // Training unit test generator $\mathcal{M}_{\text{UT}}$ (see Section 3.2)
3:      **for** $(I, C^*) \in \mathcal{D}$ **do**
4:          $\mathcal{T} \sim \mathcal{M}_{\text{UT}}(\cdot|I)$                                  ▷ Generate unit test $\mathcal{T}$ using $\mathcal{M}_{\text{UT}}$
5:          $\mathcal{C} \sim \mathcal{M}_{\text{code}}(\cdot \mid I)$                   ▷ Generate a set of code solutions $\mathcal{C}$ using $\mathcal{M}_{\text{code}}$
6:          $r_{\text{UT}} = \lambda R_{\text{disc}}(\mathcal{T}, \mathcal{C}, C^*) + (1 - \lambda)R_{\text{valid}}(\mathcal{T}, C^*, \tau)$    ▷ Compute reward for $\mathcal{T}$ (see Equation 1, 2)
7:          $\mathcal{M}_{\text{UT}} := \texttt{RL-update}(\mathcal{M}_{\text{UT}}, r_{\text{UT}})$             ▷ Update the unit test generator $\mathcal{M}_{\text{UT}}$
8:      **end for**
9:      // Training code generator $\mathcal{M}_{\text{code}}$ (see Section 3.3)
10:      **for** $(I, C^*) \in \mathcal{D}$ **do**
11:          $C \sim \mathcal{M}_{\text{code}}(\cdot|I)$                                 ▷ Generate code $C$ using $\mathcal{M}_{\text{code}}$
12:          $\mathcal{T} \sim \mathcal{M}_{\text{UT}}(\cdot \mid I)$                          ▷ Generate unit test $\mathcal{T}$ using $\mathcal{M}_{\text{UT}}$
13:          $r_{\text{code}} = R_{\text{code}}(C, \mathcal{T}, C^*)$                     ▷ Compute reward for $C$ (see Equation 4)
14:          $\mathcal{M}_{\text{code}} := \texttt{RL-update}(\mathcal{M}_{\text{code}}, r_{\text{code}})$            ▷ Update code generator $\mathcal{M}_{\text{code}}$
15:      **end for**
16: **end while**

---

- **Step 1. Training the unit test generator** $\mathcal{M}_{\text{UT}}$: For a given programming instruction $I$, we first sample multiple code solutions using the code generator LLM $\mathcal{M}_{\text{UT}}$. The unit test generator LLM $\mathcal{M}_{\text{UT}}$ is then trained to produce unit tests that (1) discriminate as many sampled code solutions from the ground-truth solution as possible, and (2) contain functionally valid test cases (see Section 3.2).

- **Step 2. Training code generator** $\mathcal{M}_{\text{code}}$: Given the programming instruction $I$, the code generator $\mathcal{M}_{\text{code}}$ is trained to generate solutions that maximize the pass rate over $\mathcal{T}$, which is generated by $\mathcal{M}_{\text{UT}}$ (see Section 3.3).

By repeating these steps, the code generator progressively learns to produce more functionally correct solutions, while the unit test generator learns to generate increasingly discriminative and high-quality unit tests.

## 3.2 TRAINING UNIT TEST GENERATOR

To train the unit test generator LLM $\mathcal{M}_{\text{UT}}$ to produce both effective and functionally valid unit tests, we introduce two reward components. The discrimination reward $R_{\text{disc}}$ serves as the primary objective, encouraging $\mathcal{M}_{\text{UT}}$ to generate unit tests that can detect faults across diverse incorrect code solutions. To simultaneously ensure the functional validity of each test case, we introduce the validity reward $R_{\text{valid}}$, which measures the proportion of functionally valid test cases among the entire test cases in the generated unit test. During training, the unit test generator $\mathcal{M}_{\text{UT}}$ is optimized with the weighted sum of $R_{\text{disc}}$ and $R_{\text{valid}}$ via RL.

**Discrimination reward** The discrimination reward evaluates how effectively the test cases in $\mathcal{T}$ discriminate LLM-generated code solutions from ground-truth code solutions. Given a unit test $\mathcal{T}$

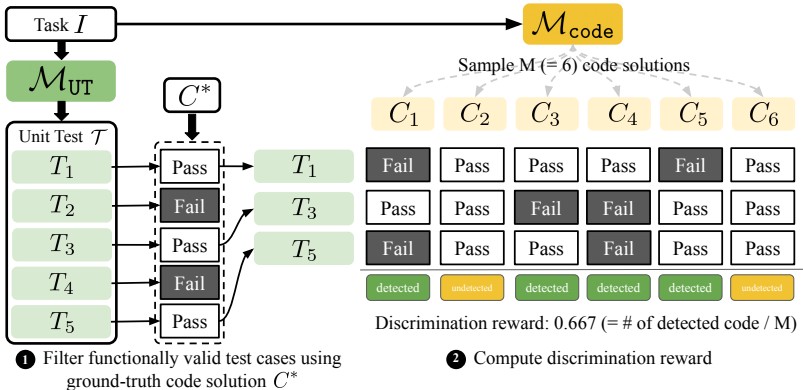

Figure 3: Overview of the process of computing discrimination reward with respect to unit test $\mathcal{T}$. First, among the 5 test cases in the unit test $\mathcal{T}$, test cases that pass under the ground-truth code $C^*$ (i.e., $T_1, T_3, T_5$) are filtered, forming a set of functionally valid test cases. Second, the discrimination reward is defined as a ratio of sampled code solutions that are detected by least one valid test case. In this figure, among 6 sampled code solutions, 4 code solutions ($C_1, C_3, C_4, C_5$) do not pass at least one valid test case, resulting in the discrimination reward of 0.667 ($= \frac{4}{6}$).

generated regarding programming instruction $I$, the discrimination reward corresponding to $\mathcal{T}$ is formally defined as follows:

$$R_{\text{disc}}(\mathcal{T}, \mathcal{C}, C^*) = \frac{1}{|\mathcal{C}|} \sum_{C \in \mathcal{C}} \left[ 1 - \prod_{T \in \mathcal{T}} (1 - \texttt{Pass}(C, T))^{\texttt{Pass}(C^*, T)} \right], \quad (1)$$

where the $C^*$ is a ground-truth code solution, $\mathcal{C}$ is a set of code solutions generated by the code generator $\mathcal{M}_{\texttt{code}}$, and $\texttt{Pass}(C, T)$ is an indicator function that returns 1 if the code $C$ passes the test case $T$, and 0 otherwise.[0] To compute the discrimination reward regarding unit test $\mathcal{T}$, as illustrated in Figure 3, we first filter out functionally invalid test cases from the generated unit test $\mathcal{T}$, ensuring that invalid test cases do not affect the discrimination reward. We then leverage a set of code solutions $\mathcal{C}$ generated by $\mathcal{M}_{\texttt{code}}$ (i.e., $\mathcal{C} \sim \mathcal{M}_{\texttt{code}}(\cdot \mid I)$), where the discrimination reward is computed as a fraction of the code solutions that do not pass the filtered unit test.

**Validity reward** The validity reward evaluates whether each test case in the generated unit test $\mathcal{T}$ is functionally valid (i.e., correctly maps the input and expected output of the code solution), and is formally defined as follows:

$$R_{\text{valid}}(\mathcal{T}, C^*, \tau) = \frac{\sum_{T \in \mathcal{T}} \texttt{Pass}(C^*, T)}{\max(|\mathcal{T}|, \tau)}, \quad (2)$$

where $\tau$ is a hyperparameter to adjust the desired minimum number of test cases in a single generated unit test. We apply clipping to the denominator in Equation 2 as $\max(|\mathcal{T}|, \tau)$, in order to prevent unit tests with only a small number of trivial test cases from receiving an inappropriately high validity reward. To be specific, if the $R_{\text{valid}}$ is simply defined as a fraction of valid test cases among total test cases (i.e., $R_{\text{valid}}(\mathcal{T}, C^*) = \sum_{T \in \mathcal{T}} \texttt{Pass}(C^*, T)/|\mathcal{T}|$), unit tests with an extremely small number of test cases (e.g., unit tests with a single trivial test case) acquire high validity reward, which is undesirable. To mitigate this problem, we clip the denominator as $\max(|\mathcal{T}|, \tau)$. This ensures that unit tests with fewer than $\tau$ test cases receive a validity reward proportional to the absolute number of valid test cases, thereby preventing the unit tests with a small number of trivial test cases from receiving high validity reward.

**Update unit test generator LLM** In order to guide the unit test generator LLM to produce unit test comprising test cases that are both (1) highly discriminative and (2) functionally valid, we define a training reward as a weighted sum of the discrimination reward $R_{\text{disc}}$ and the validity reward $R_{\text{valid}}$:

$$r_{\text{UT}} = \lambda R_{\text{disc}}(\mathcal{T}, \mathcal{C}, C^*) + (1 - \lambda) R_{\text{valid}}(\mathcal{T}, C^*, \tau), \quad (3)$$

where the $\lambda$ is a hyperparameter that weights the discrimination reward.

---

[0]See Appendix A.3 for details about the implementation of $\texttt{Pass}$ function.

### 3.3 TRAINING CODE GENERATOR

After training the unit test generator LLM $\mathcal{M}_{\mathrm{UT}}$, we train the code generator LLM $\mathcal{M}_{\mathrm{code}}$ via RL. Regarding a code $C$ generated by $\mathcal{M}_{\mathrm{code}}$, the training reward is formally defined as follows:

$$R_{\mathrm{code}}(C, \mathcal{T}, C^*) = \frac{\sum_{T \in \mathcal{T}} \mathrm{Pass}(C, T) \cdot \mathrm{Pass}(C^*, T)}{\sum_{T \in \mathcal{T}} \mathrm{Pass}(C^*, T)}, \tag{4}$$

where $C^*$ is a ground-truth code solution. In detail, we first filter out functionally invalid test cases (i.e., test cases that do not pass under the ground-truth code solution $C^*$) from $\mathcal{T}$, because such invalid test cases lead to faulty evaluation of the code. We then measure the proportion of the functionally valid test cases that are passed by the generated code $C$. Based on the reward design, we optimize $\mathcal{M}_{\mathrm{code}}$ to produce code solutions that pass unit tests generated by $\mathcal{M}_{\mathrm{UT}}$.

## 4 EXPERIMENTS

We design our experiments to investigate the following questions:

- **RQ1**. Is UTRL effective at training LLMs to generate high-quality unit tests? (see Section 4.2)

- **RQ2**. Is UTRL more effective than supervised learning-based approaches? (see Section 4.3)

- **RQ3**. Is UTRL effective at improving code generation? (see Section 4.4)

- **RQ4**. Does iterative training of UTRL enable continuous improvement of unit test generation capability? (see Section 4.5)

### 4.1 EXPERIMENTAL SETUP

In this section, we describe training details, baselines, and evaluation protocols for our experiments.

**Training details**  As the base model for both the code generator and unit test generator, we use Qwen3-4B (Yang et al., 2025). For RL training, we adopt Grouped Relative Policy Optimization (GRPO; Shao et al. 2024 (see Appendix A.1 for details) and use 15,249 programming instruction-code solution pairs from the TACO dataset (Li et al., 2023). Further details about training and prompts are described in Appendix A.3 and A.5.

**Baselines**  We compare our method to various baselines for unit test generation. Implementation details for the baselines are provided in Appendix A.3.

- **LLM baselines without fine-tuning**: We use 4 open-source LLMs (Qwen3-4B, Qwen3-8B, Qwen3-14B, Qwen3-32B, Deepseek-Coder-V2-Lite; Qwen et al. 2025; Zhu et al. 2024) and 2 closed LLMs (GPT-4o, GPT-4.1; Hurst et al. 2024; OpenAI 2025) as baselines[1].

- **Supervised fine-tuning (SFT)**: We fine-tune Qwen3-4B with instruction–unit test pairs through supervised learning. For this purpose, we use high-quality unit tests from the TACO training dataset $\mathcal{D}_{\mathrm{UT}}$. To investigate whether reasoning signals can improve unit test generation, we construct a reasoning-augmented dataset $\mathcal{D}_{\mathrm{reason+UT}}$ containing unit tests paired with their corresponding reasoning. Specifically, we first generate reasoning and unit tests for each training task using Gemini-2.5-flash (Google Deepmind, 2025).[2] We then filter functionally valid reasoning–test case pairs and use them as target labels for SFT training.

- **CURE**: We also consider CURE (Wang et al., 2025a), an RL algorithm that fine-tunes an LLM to perform both test case generation and code generation using a dataset of instruction–unit test pairs. Specifically, we evaluate unit tests generated by ReasonFlux-Coder-7B(Wang et al., 2025a), which is based on Qwen2.5-7B-Instruct (Yang et al., 2025) fine-tuned with CURE.

- **Ground-truth (GT) unit tests**: We also compare against the unit tests provided in the benchamrk datasets (Li et al., 2023; Jain et al., 2024b), which we consider as ground truth (GT) and regard as an upper bound since they are rigorously verified and consist of a large number of high-quality test cases covering diverse edge cases.

---

[1]We note that 4 open-source LLMs used as baselines are instruction-finetuned models.

[2]Instead of attaching reasoning to the original unit tests in TACO, which led to frequent hallucinations and inconsistencies, we regenerate both reasoning and unit tests together.

| Unit test generated by | Code LLM: Qwen3-8B (32 shot) | | | | Code LLM: Qwen3-14B (32 shot) | | | |
|---|---|---|---|---|---|---|---|---|
| | Score | $\Delta$Score | Acc (%) | $\Delta$Acc (%p) | Score | $\Delta$Score | Acc (%) | $\Delta$Acc (%p) |
| Qwen3-4B | 0.430 | +0.084 | 9.8 | +1.8 | 0.459 | +0.057 | 13.4 | +2.7 |
| Qwen3-8B | 0.435 | +0.089 | 9.7 | +1.7 | 0.470 | +0.070 | 12.6 | +1.9 |
| Qwen3-14B | 0.460 | +0.114 | 9.8 | +1.8 | 0.483 | +0.083 | 11.6 | +0.9 |
| Qwen3-32B | 0.484 | +0.138 | 11.7 | +3.7 | 0.514 | +0.114 | 13.7 | +3.0 |
| Deepseek-Coder-V2-Lite | 0.433 | +0.087 | 9.5 | +1.5 | 0.452 | +0.050 | 11.9 | +1.2 |
| GPT-4o | 0.474 | +0.128 | 10.9 | +2.9 | 0.502 | +0.100 | 11.5 | +0.8 |
| GPT-4.1 | 0.537 | +0.191 | 13.3 | +5.3 | 0.572 | +0.170 | 15.1 | +4.4 |
| Qwen3-4B + SFT w/ $\mathcal{D}_{UT}$ | 0.458 | +0.112 | 11.7 | +3.7 | 0.491 | +0.091 | 14.0 | +3.3 |
| Qwen3-4B + UTRL (ours) | 0.578 | +0.232 | 14.9 | +6.9 | 0.610 | +0.208 | 17.3 | +6.6 |
| Qwen3-14B + UTRL (ours) | **0.599** | **+0.253** | **15.0** | **+7.0** | **0.627** | **+0.225** | **17.7** | **+7.0** |
| GT (upperbound) | 0.668 | +0.304 | 20.3 | +13.1 | 0.699 | +0.278 | 24.2 | +13.7 |

Table 1: Best-of-$N$ improvement achieved by Qwen3-4B and Qwen3-14B trained via UTRL, compared against baselines. We report code score (Score) and code accuracy (Acc) of the best-of-$N$ selected code solution, and also report the increment of each metric compared to code generated without the best-of-$N$ sampling (i.e., $\Delta$Score and $\Delta$Acc). This result demonstrates effectiveness of UTRL in training LLMs to produce unit tests with highly discriminative test cases.

**Evaluation protocols** For evaluation, we utilize rigorously verified 945 competitive programming tasks provided in the TACO evaluation dataset (Li et al., 2023) and 511 programming tasks provided in the LiveCodeBench-v2 (Jain et al., 2024b). To measure the quality of the generated unit tests, we propose the following evaluation schemes:

- **Best-of-$N$ improvement**: To examine whether the generated unit tests include high-quality test cases, we perform best-of-$N$ sampling for code generation, using the generated unit tests as evaluators. Specifically, we generate $N$ candidate solutions from code LLMs and select the one that passes the largest number of generated unit tests. To evaluate the quality of the selected code, we then measure two metrics: (1) code score, defined as the fraction of test cases in a GT unit test (i,e., high-quality unit tests from evaluation dataset) that the code solution passes, and (2) code accuracy, which indicates whether the solution passes all test cases in the GT unit test.

- **Unit test fidelity**: To quantify how closely a generated unit test approximates the evaluation induced by the GT unit test, we introduce a metric called *unit test fidelity*. For each task, we first sample multiple code solutions from LLMs. We then evaluate these solutions twice: once using the generated unit test and once using the GT unit test. This yields two code score vectors, and we compute Spearman's correlation between them. A high correlation indicates that the generated unit test closely replicates the evaluation induced by the comprehensive GT unit test.

We provide more details about the evaluation metrics in Appendix A.4.

## 4.2 QUALITY OF UNIT TESTS

In this section, we demonstrate the effectiveness of UTRL in training LLMs to generate high-quality unit tests, compared with strong baselines. We first assess the quality of the generated unit tests using Best-of-$N$ improvement and unit test fidelity. We then compare UTRL against a recent RL-based baseline, CURE.

**Best-of-$N$ improvement** Table 1 reports the best-of-$N$ improvement achieved by Qwen3-4B and Qwen3-14B trained with UTRL, compared against several baselines. When used as code evaluators for best-of-32 sampling with Qwen3-8B and Qwen3-14B code generators, unit tests produced by Qwen3-4B trained with UTRL yield code accuracies of 14.9% and 17.3%, respectively. For comparison, Qwen3-4B trained with SFT achieves only 11.7% and 14.0%. Moreover, Qwen3-4B trained with UTRL outperforms frontier proprietary models such as GPT-4.1 and GPT-4o, underscoring the effectiveness of UTRL in training LLMs to generate high-quality unit tests. With increased model capacity, Qwen3-14B trained with UTRL attains stronger performance, reaching code accuracies of 15.0% and 17.7%. Notably, UTRL is effective even when paired with closed-source code generators such as GPT-4o (see Table 8 in Appendix A.2 for supporting results).

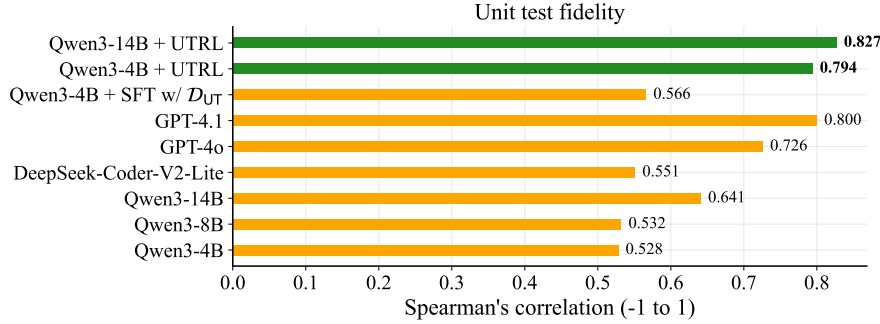

Figure 4: Fidelity of unit tests generated by Qwen3-4B and Qwen3-14B trained with UTRL, compared against baselines. Both model trained via UTRL achieves the unit test fidelity higher than baselines, demonstrating the effectiveness of UTRL in training LLMs to generate unit tests that closely approximates the code evaluation induced by the GT unit tests.

| Unit test generated by | Code LLM: Qwen3-8B (32 shot) | | | | Code LLM: Qwen3-14B (32 shot) | | | |
|---|---|---|---|---|---|---|---|---|
| | Score | $\Delta$Score (%) | Acc (%) | $\Delta$Acc (%p) | Score | $\Delta$Score | Acc (%) | $\Delta$Acc (%p) |
| CURE | 0.446 | +0.100 | 9.7 | +1.7 | 0.483 | +0.083 | 12.7 | +2.0 |
| UTRL$^{\dagger}$ (ours) | **0.521** | **+0.175** | **14.1** | **+6.1** | **0.562** | **+0.162** | **15.9** | **+5.2** |
| GT (upperbound) | 0.650 | +0.304 | 21.0 | +13.1 | 0.680 | +0.278 | 24.4 | +13.7 |

Table 2: Best-of-$N$ improvement induced by UTRL and CURE. For UTRL, we train Qwen2.5-7B-Instruct using 4.5K programming tasks in CodeContest dataset (Li et al., 2022) following the training setup of CURE, which we denote as UTRL$^{\dagger}$.

**Unit test fidelity** Figure 4 presents the fidelity of unit tests generated by Qwen3-4B trained with UTRL, compared against several baselines. Qwen3-14B and Qwen3-4B trained with UTRL achieves a Spearman's correlation of 0.827 and 0.794, where Qwen3-14B trained via UTRL outperforms GPT-4.1, demonstrating the effectiveness of UTRL in training LLMs to generate unit tests that induce code evaluations consistent with the GT unit tests. Interestingly, while SFT with $\mathcal{D}_{UT}$ provides substantial gains in best-of-$N$ improvement relative to the base Qwen3-4B, it yields only marginal improvements in unit test fidelity. We conjecture that this is because SFT with $\mathcal{D}_{UT}$ trains the LLM to generate unit tests without step-by-step reasoning, making it difficult for the model to produce logically structured test suites (i.e., ranging from basic test cases to highly discriminative edge cases), which are essential for achieving high unit test fidelity.

**Comparison to CURE** Table 2 and Figure 6 report the best-of-$N$ improvement and unit test fidelity achieved by Qwen2.5-7B-Instruct trained with UTRL versus the same model trained with CURE, an RL baseline based on instruction–unit test pairs. When used as a code evaluator for best-of-32 sampling with Qwen3-8B and Qwen3-14B code generators, UTRL achieves code accuracies of 14.1% and 15.9%, which are 4.4% and 3.2% higher than those obtained with CURE. Moreover, UTRL achieves a unit test fidelity of 0.593, surpassing CURE. These results show that UTRL achieves superior performance despite relying only on instruction–code pairs, in contrast to CURE requiring instruction–unit test pairs.

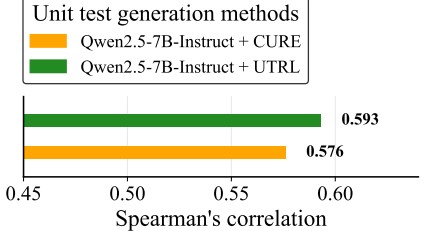

Figure 6: Unit test fidelity of UTRL compared against CURE. Qwen2.5-7B-Instruct trained via UTRL achieves unit test fidelity higher than CURE, demonstrating superiority of UTRL over CURE.

## 4.3 SUPERIORITY OF UTRL OVER SUPERVISED LEARNING APPROACHES

To investigate whether training LLMs with reasoning-annotated unit tests via SFT can improve unit test generation, we also evaluate SFT with $\mathcal{D}_{reason+UT}$ and compare it with UTRL. Figure 5 presents

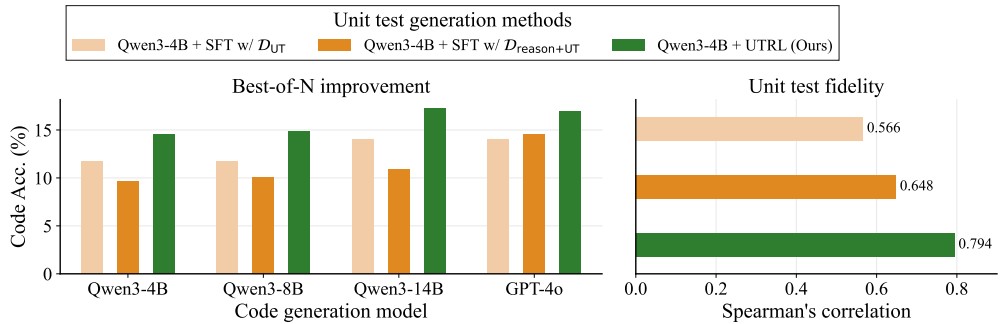

Figure 5: Best-of-$N$ improvmenet (**left**) and unit test fidelity (**right**) achieved by UTRL, compared against SFT baselines (SFT w/ $\mathcal{D}_{\text{UT}}$, SFT w/ $\mathcal{D}_{\text{reason+UT}}$).

best-of-$N$ improvement (left) and unit test fidelity (right) for Qwen3-4B trained with SFT baselines versus the same model trained with UTRL. While SFT + $\mathcal{D}_{\text{reason+UT}}$ achieves an improved unit test fidelity score of 0.604, higher than the SFT + $\mathcal{D}_{\text{UT}}$, it falls behind 0.741 achieved by UTRL. Additionally, in best-of-$N$ improvement, SFT + $\mathcal{D}_{\text{reason+UT}}$ shows performance even inferior to SFT + $\mathcal{D}_{\text{UT}}$. We conjecture that this is because the training dataset $\mathcal{D}_{\text{reason+UT}}$ comprises synthetic test cases generated by Gemini-2.5-Flash rather than highly discriminative ground-truth unit tests that constitute $\mathcal{D}_{\text{UT}}$. Hence, the model trained with $\mathcal{D}_{\text{reason+UT}}$ generates relatively less discriminative unit tests compared to the same model trained with $\mathcal{D}_{\text{UT}}$, which results in inferior best-of-N improvement, since Best-of-N improvement is largely determined by the presence of highly discriminative test cases. Although SFT baselines directly optimize LLMs using unit test labels annotated by humans or more capable teacher models, they show limited generalization to evaluation tasks compared with UTRL. This observation is consistent with prior findings that SFT tends to memorize the training distribution, whereas RL promotes better generalization in reasoning intensive tasks (Chu et al., 2025). Moreover, since collecting high-quality unit test annotations for SFT is costly, UTRL offers a more practical approach for training LLMs for unit test generation.

## 4.4 EFFECTIVENESS OF UTRL IN TRAINING CODE GENERATOR

In UTRL, the code generator is adversarially trained against the unit test generator to produce code solutions that maximize the pass rate over the generated unit tests. To evaluate the effectiveness of UTRL for training the code generator, we measure the pass@1 code accuracy of code solutions generated by Qwen3-4B trained via UTRL, and compare it against two code generator training baselines: (1) RL using rewards defined as the pass rate over unit tests generated by GPT-4o, and (2) SFT using ground-truth code solution provided in the training dataset. Moreover, we also evaluate Qwen3-4B trained via RL using rewards defined as the pass rate over GT unit tests provided in the TACO dataset, which we regard as our upperbound. Figure 7 shows pass@1 code accuracy averaged over 945 evaluation tasks. The code generator trained with UTRL achieves a code accuracy of 15.3%, which is remarkably higher than the two baselines, and even comparable to 15.9% achieved by the code generator trained to maximize pass rate over the GT unit tests. While training the code generator to maximize the pass rate over GPT-4o-generated unit tests improved the pass@1 accuracy to some extent, the code accuracy saturates under 12%, and we observe that directly fine tuning code generator with ground-truth code solutions via SFT severely degrades the performance on unseen evaluation tasks, resulting in merely 3.6% pass@1 code accuracy. These results indicate the effectiveness of UTRL in training the code generator, especially when high-quality ground-truth unit tests are infeasible.

## 4.5 EFFECT OF ITERATIVE TRAINING

We examine whether iterative training enables the code generator to produce near-perfect code solution, thereby providing increasingly challenging discrimination tasks for the unit test generator, and whether the generator improves by learning to discriminate the near-perfect code from ground-truth code. Figure 8 shows discrimination rewards averaged over evaluation tasks. In the first iteration, the unit test generator is trained against code produced by the initial code generator, and the discrimination reward saturates after about 50 steps. At this point, we stop training and update the code generator via RL to maximize its pass rate against the current unit tests. When training resumes in iteration 2, the discrimination reward drops from 0.626 to 0.375, indicating that the updated code

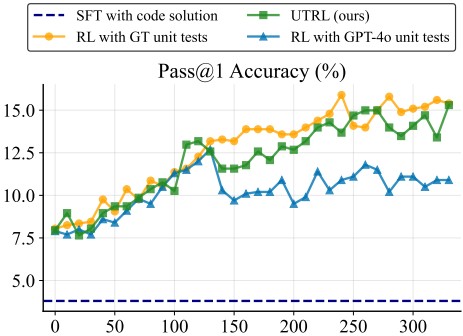

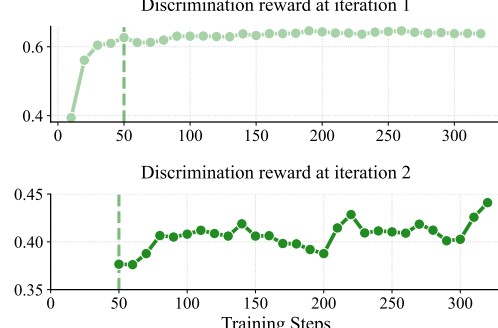

Figure 7: Pass@1 accuracy of code generators on the TACO evaluation tasks. Qwen3-4B code generator trained with UTRL outperforms the same model trained with GPT-4o-generated unit tests and the same model trained via SFT in terms of pass@1 accuracy.

Figure 8: Discrimination reward evaluated at training iterations 1 and 2. During iteration 1 training, the discrimination reward saturated after 50 steps with only about a gain of 0.02, while in iteration 2, it exhibited an improvement of 0.07.

generator now produces code solutions that are harder to distinguish from ground-truth, thereby creating a more challenging discrimination task for the unit test generator. As training continues in iteration 2, the unit test generator improves the reward from 0.375 to 0.447. Correspondingly, as shown in Figure 9, the unit test generator at iteration 2 produces unit tests that yield superior best-of-$N$ performance compared to iteration 1, even outperforming GPT-4.1. Furthermore, we extend the training to a third iteration, but we did not observe notable improvement in discrimination reward. While we conducted these experiments using Qwen3-4B as the unit test generator, we believe that employing a larger-capacity model as a unit test generator would likely allow the adversarial process to sustain meaningful improvements over more iterations.

## 5 CONCLUSION

In this work, we present UTRL, an adversarial reinforcement learning framework for training LLM as unit test generator. Our approach alternates between training the unit test generator and a code generator, enabling the unit test generator to produce unit tests that can distinguish between near-correct code solutions and the ground-truth code implementation. Through extensive experiments and analysis, we demonstrate the superiority of UTRL over recent baselines and supervised learning-based methods, even outperforming frontier models like GPT-4.1. We hope this work paves the way for future research on training algorithm for unit test generation.

### ETHICS STATEMENT

We introduce UTRL, an RL framework for training LLMs for unit test generation. As unit tests enable systematic and interpretable verification over source code written by LLMs, we believe the improvement in automated unit test generation will contribute to more reliable and safe LLM-based code generation and agentic software engineering.

### REPRODUCIBILITY STATEMENT

For the reproducibility of our results, we have provided a detailed description of our experimental setups and prompts in Section A.3, A.4, and A.5. Additionally, to further facilitate reproduction, we will open-source the training dataset, model checkpoint, and source code.

### ACKNOWLEDGEMENT

This research was conducted as part of the Sovereign AI Foundation Model Project(Data Track), organized by the Ministry of Science and ICT(MSIT) and supported by the National Information Society Agency(NIA), S.Korea. (Grant No. 2026-AIData-WII01). This work was also supported by Institute for Information communications Technology Planning Evaluation(IITP) grant funded

by the Korea government(MSIT) (RS-2019-II190075, Artificial Intelligence Graduate School Program(KAIST)). This work was also supported by the National Research Foundation of Korea(NRF) grant funded by the Korea government(MSIT) (No. RS-2024-00414822).

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

## A  APPENDIX

### A.1  GROUPED RELATIVE POLICY OPTIMIZATION

In this section, we describe the RL algorithm we used for training the LLMs in our experiments.

**Proximial Policy Optimization**  Proximal Policy Optimization (PPO) (Schulman et al., 2017), an actor-critic RL algorithm, is widely used for training LLMs via RL. PPO updates the LLM policy $\pi_\theta$ by maximizing the clipped surrogate objective described as follows:

$$\mathcal{J}_{\text{PPO}}(\theta) = \mathbb{E}_{o \sim \pi_{\theta_{\text{old}}}(\cdot|x)} \left[ \sum_{t=1}^{|o|} \min \left( \frac{\pi_\theta(o_t|x, o_{<t})}{\pi_{\theta_{\text{old}}}(o_t|x, o_{<t})} A(x, o_t), \right. \right.$$

$$\left. \left. \text{clip} \left( \frac{\pi_\theta(o_t|x, o_{<t})}{\pi_{\theta_{\text{old}}}(o_t|x, o_{<t})}, 1 - \epsilon, 1 + \epsilon \right) A(x, o_t) \right) \right], \tag{5}$$

where $\pi_\theta$ is a policy under the training, $x$ denotes a prompt, $o_{<t}$ is a completion up to $t$-th token, $o_t$ is a $t$-th token generated by the LLM $\pi_{\theta_{\text{old}}}$, and $\epsilon$ is a hyperparameter for clipping the importance sampling ratio. Here, the computation of advantage $A(x, o_t)$ requires learning a separate value function $V_\psi$. Usually, such a value function is parameterized by LLM of size comparable to the LLM policy under training, thereby requiring significant memory and computational cost.

**Grouped Relative Policy Optimization**  In order to bypass the cost of learning the value function, Grouped Relative Policy Optimization (GRPO) has been proposed as a variant of PPO (Shao et al., 2024). In order to compute the advantage, GRPO samples a group of $G$ outputs $\{o^{(i)}\}_{i=1}^G$ from the old LLM policy $\pi_{\theta_{\text{old}}}$ for input prompt $x$, and computes scalar rewards $\{r_i = R(x, o^{(i)})\}_{i=1}^G$. These rewards are then normalized by subtracting the group mean and dividing by the group standard deviation. The resulting normalized score $A_i = \frac{r_i - \text{mean}(r)}{\text{std}(r)}$ is broadcast to all tokens $t$ in the output $o^{(i)}$, such that $A(x, o_t^{(i)}) = A_i$. This shared advantage is then used to weight the token-level policy gradients, allowing for effective optimization without a learned value function. Formally, GRPO optimizes the following objective:

$$\mathcal{J}_{\text{GRPO}}(\theta) = \mathbb{E}_{\{o^{(i)}\}_{i=1}^G \sim \pi_{\theta_{\text{old}}}(\cdot|x)} \left[ \frac{1}{G} \sum_{i=1}^G \frac{1}{|o^{(i)}|} \sum_{t=1}^{|o^{(i)}|} \min \left( \frac{\pi_\theta(o_t^{(i)}|x, o_{<t}^{(i)})}{\pi_{\theta_{\text{old}}}(o_t^{(i)}|x, o_{<t}^{(i)})} A_i, \right. \right.$$

$$\left. \left. \text{clip} \left( \frac{\pi_\theta(o_t^{(i)}|x, o_{<t}^{(i)})}{\pi_{\theta_{\text{old}}}(o_t^{(i)}|x, o_{<t}^{(i)})}, 1 - \epsilon, 1 + \epsilon \right) A_i \right) \right], \tag{6}$$

where $A_i = \frac{r_i - \text{mean}(r)}{\text{std}(r)}$ is the normalized advantage shared across all tokens in output $o^{(i)}$.

### A.2  ADDITIONAL EXPERIMENTS

In this section, we provide supplementary experiments to thoroughly validate each component of UTRL, and demonstrate further advantage of UTRL.

#### A.2.1  EVALUATION ON LIVECODEBENCH

To evaluate whether the unit test generator LLM trained via UTRL generalizes to arbitrary diverse programming tasks, we additionally measure Best-of-N improvement and unit test fidelity of UTRL on 511 programming tasks in LiveCodeBench (Jain et al., 2024b). As shown in the Table 3, Qwen3-4B trained via UTRL achieves best-of-N improvement higher than GPT-4.1, and Qwen2.5-7B-Instruct trained via UTRL outperforms the same model trained via CURE. This results underscore that the effectiveness of unit test generator trained via UTRL well generalizes to diverse programming tasks.

#### A.2.2  COMPARISON TO BEST-OF-N DISTILLATION

As an additional baseline for training unit test generator, we compare UTRL with best-of-N distillation approach. For Best-of-N distillation, we sample 8 unit tests for each training instance using Qwen3-4B, select one with maximum discrimination reward, and finetune the Qwen3-4B with the selected unit test via SFT. As shown in the Table 4, although the best-of-N distillation approach yields performance comparable to SFT w/ $\mathcal{D}_{\text{UT}}$ baseline even without requiring ground-truth unit tests for training, UTRL achieves superior performance.

| Unit test generated by | Code LLM: Qwen3-4B | | Code LLM: Qwen3-8B | |
|---|---|---|---|---|
| | Score | Acc (%) | Score | Acc (%) |
| Qwen3-4B | 0.704 | 55.2 | 0.714 | 55.4 |
| Qwen3-4B + UTRL | **0.744** | **59.9** | **0.752** | **59.3** |
| Qwen2.5-7B | 0.607 | 44.6 | 0.617 | 44.0 |
| Qwen2.5-7B+CURE | 0.660 | 48.2 | 0.671 | 47.7 |
| Qwen2.5-7B+UTRL [†] | 0.686 | 54.4 | 0.693 | 52.4 |
| GPT-4.1 | 0.732 | 58.3 | 0.735 | 58.3 |

Table 3: Best-of-N improvement evaluation on LiveCodeBench. A unit test generator trained via UTRL also generalizes to unseen tasks in LiveCodeBench, outperforming CURE. Moreover, Qwen3-4B trained via UTRL achieves performance superior to GPT-4.1.

| Unit test generated by | Code LLM: Qwen3-4B | | Code LLM: Qwen3-8B | |
|---|---|---|---|---|
| | Score | Acc (%) | Score | Acc (%) |
| Qwen3-4B + SFT w/ $\mathcal{D}_{\text{UT}}$ | 0.368 | 7.8 | 0.380 | 7.7 |
| Qwen3-4B + best-of-N distil. | 0.368 | 7.8 | 0.380 | 7.7 |
| Qwen3-4B + UTRL | **0.475** | **11.6** | **0.494** | **11.9** |

Table 4: Comparison to best-of-N distllation approach for training unit test generator.

### A.2.3 EXPERIMENT ON DIFFERENT LLM

Although we mainly train Qwen3-4B and Qwen3-14B via UTRL, we additionally show effectiveness of UTRL in training arbitrary LLMs. To this end, we train Llama3.1-8B-Instruct via UTRL for 50 steps and evaluate best-of-N improvement of the unit tests generated by the resulting unit test generator. As shown in the Table 7, UTRL effectively enhances the unit test generation capability when applied to the Llama3.1-8B-Instruct, improving achieving best-of-N improvement of 11.75% on average, which is remarkably higher than Llama-3.1-8B-Instruct model.

### A.2.4 ABLATION ON WEIGHTING TERM FOR DISCRIMINATION REWARD

We conduct ablation study on hyperparameter $\lambda$, which controls the weight of the discrimination reward for training unit test generator LLM. We sweep over $\lambda \in (0.0, 0.85, 1.0)$. As shown in the Table 6 and Table 7, setting $\lambda = 0.0$ leads to improved validity (proportion of the functionally correct test cases among entire test cases in the unit test), while resulting in lower Best-of-N improvement, since the model is guided to generate test cases that are merely valid but indifferent to discriminativeness. Conversely, setting $\lambda = 1.0$ results in a substantial fraction of invalid test cases, which also degrades unit test quality measured by Best-of-N improvement. Setting $\lambda = 0.85$ achieves the best unit test quality, balancing validity and discriminativeness of unit tests. These results imply that both discrimination reward and validity reward are essential, and should be balanced appropriately to produce high-quality unit tests.

### A.2.5 ABLATION ON VALIDITY REWARD

In order to demonstrate the effectiveness of the validity reward and its design choice (see Equation 2), we conduct an ablation study comparing the following variants of the UTRL, (1) UTRL trained without validity reward (i.e., w/o $R_{\text{valid}}$), and (2) UTRL with validity reward but without clipped normalization in validity reward term, where the validity reward is defined as a ratio of valid test cases among the generated test cases (i.e., w/o clipping). Figure 10 shows the number of test cases and the ratio of valid test cases in the generated unit test over training steps. UTRL without $R_{\text{valid}}$ results in the generation of unit tests with more than 50% of invalid test cases, while UTRL results in unit tests containing less than 35% of invalid test cases. Furthermore, in the case of UTRL without clipping, we observe that the model collapses to generate an extremely small number of trivial test cases in order to maximize the ratio of the valid test cases, hindering the learning progress toward generating comprehensive unit tests. These results demonstrates the effectiveness of the design choice of UTRL.

| Unit test generated by | Code LLM: Qwen3-4B | | Code LLM: Qwen3-8B | |
|---|---|---|---|---|
| | Score | Acc (%) | Score | Acc (%) |
| Llama-3.1-8B-Instruct | 0.368 | 7.8 | 0.380 | 7.7 |
| Llama-3.1-8B-Instruct + UTRL | **0.475** | **11.6** | **0.494** | **11.9** |

Table 5: Performance of Llama-3.1-8B-Instruct trained via UTRL. UTRL effectively improve the unit test generation capability of Llama-3.1-8B-Instruct, which is measured by best-of-N improvement.

| | Code LLM: Qwen3-4B | | Code LLM: Qwen3-8B | |
|---|---|---|---|---|
| | Score | Acc (%) | Score | Acc (%) |
| $\lambda = 0$ | 0.509 | 12.4 | 0.524 | 13.1 |
| $\lambda = 0.85$ | **0.514** | **13.7** | **0.534** | **13.8** |
| $\lambda = 1.0$ | 0.485 | 12.3 | 0.492 | 11.5 |

Table 6: Best-of-N improvement according to weighting factor for discrimination reward $\lambda$. Omitting validity reward ($\lambda = 0$) degrades the quality of the unit tests owing to the functionally incorrect test cases occupying more than half of the generated unit test.

### A.2.6 Best-of-N improvement measured with additional code generators

Table 8 presents Best-of-N improvement measured when Qwen3-4B and GPT-4o is used for code generator LLM. In this evaluation, UTRL still achieves the highest code accuracy and code score, demonstrating that LLMs trained via UTRL produces high-quality unit tests that can effectively evaluate code generated by various code LLMs.

### A.3 Implementation details

In this section, we provide implementation details of UTRL and other baselines in our experiments. Hyperparameters are described in Table 9.

**Training details of UTRL** For training, we utilize 15,249 programming task instances provided in training split of TACO dataset, where we exclude instances without a ground-truth code solution and those whose unit test annotation do not follow `stdio` format (e.g., functional assertions), and 1,000 programming task instances for validation. In our experiment, we train instruction-finetuned Qwen3-4B and Qwen3-14B via UTRL. For Qwen3-4B experiment, we train the unit test generator for 50 steps in the first iteration, and train the code generator for 330 steps, and then continually train the unit test generator for 170 steps in the second iteration. For Qwen3-14B experiment, we train the unit test generator for 100 steps at the first iteration, without further iteration. We use weighted sum of discrimination reward and validity reward (see Equation 3) measured on the 1,000 validation tasks as a validation metric to choose training steps. For the number of LLM-generated code samples to define a discrimination reward, we set the value as 8. For $\tau$, a hyperparameter to adjust the desired minimum number of the test cases in the unit test, we set the value as 12. Additionally, for $\lambda$, a weighting factor for the discrimination reward, we set this value as 0.85. Moreover, in order to ensure fair comparison with CURE, we also implement UTRL $^{\dagger}$ by training Qwen2.5-7B-Instruct via UTRL for 100 steps using the 4.5K instruction-code solution pairs in CodeContests dataset, following the experimental setup of CURE. For UTRL $^{\dagger}$, we trained the model only for a single iteration.

**Implementation of `Pass` function** We describe implementation details about the `Pass` function, which is core component of reward computation. Given a code $C$ and a test case $T$, we consider `Pass`$(C, T)$ is 1 (i.e., code $C$ passes test case $T$) if (1) the code $C$ builds successfully without any syntax error, (2) the execution of test case $T$ regarding the code $C$ does not return any error (e.g., assertion error), and (3) the execution of test case $T$ regarding the code $C$ runs under 10 seconds. Otherwise, we consider `Pass`$(C, T)$ is 0 (i.e., code $C$ does not pass test case $T$).

**Details of baselines**

- **SFT with $\mathcal{D}_{\text{UT}}$**: As a training dataset, we use the 15,249 training task instances in TACO dataset, exactly same as UTRL, and we fine-tune Qwen3-4B. To label the unit test for SFT, we randomly

|  | validity ratio (%) |
|---|---|
| $\lambda = 0$ | 64.3 |
| $\lambda = 0.85$ | 57.2 |
| $\lambda = 1.0$ | 49.7 |

Table 7: Ratio of valid test cases according to weighting factor for discrimination reward $\lambda$.

| Unit test generated by | Qwen3-4B | | | | GPT-4o | | | |
|---|---|---|---|---|---|---|---|---|
|  | Score | $\Delta$Score | Acc (%) | $\Delta$Acc (%p) | Score | $\Delta$Score | Acc (%) | $\Delta$Acc (%p) |
| Qwen3-4B | 0.430 | +0.084 | 9.8 | +1.8 | 0.459 | +0.057 | 13.4 | +2.7 |
| Qwen3-8B | 0.435 | +0.089 | 9.7 | +1.7 | 0.470 | +0.070 | 12.6 | +1.9 |
| Qwen3-14B | 0.461 | +0.115 | 9.7 | +1.7 | 0.475 | +0.075 | 13.3 | +2.6 |
| DeepSeek-Coder-Lite-V2 | 0.407 | +0.061 | 9.0 | +1.0 | 0.458 | +0.056 | 12.9 | +2.2 |
| GPT-4o | 0.463 | +0.111 | 9.6 | +2.6 | 0.511 | +0.094 | 13.8 | +1.4 |
| GPT-4.1 | 0.529 | +0.182 | 12.9 | +6.2 | 0.547 | +0.148 | 15.6 | +4.6 |
| Qwen3-4B + SFT w/ $\mathcal{D}_{UT}$ | 0.457 | +0.127 | 11.7 | +4.5 | 0.437 | +0.239 | 14.0 | +7.5 |
| Qwen2.5-7B + CURE | 0.421 | +0.092 | 10.2 | +3.0 | 0.459 | +0.261 | 13.8 | +7.3 |
| Qwen2.5-7B + UTRL [†] | 0.502 | +0.173 | 14.0 | +6.8 | 0.526 | +0.328 | 16.0 | +9.5 |
| Qwen3-4B + UTRL | 0.555 | +0.207 | 14.5 | +7.0 | 0.568 | +0.353 | 16.9 | +10.2 |
| Qwen3-14B + UTRL | **0.579** | **+0.219** | **15.8** | **+7.8** | **0.578** | **+0.359** | **17.0** | **+11.0** |
| GT (upperbound) | 0.630 | +0.284 | 20.2 | +13.0 | 0.599 | +0.197 | 21.7 | +11.0 |

Table 8: Best-of-N improvement measured on TACO evaluation set using Qwen3-4B and GPT-4o as code generator LLM. UTRL shows the highest best-of-N improvement compared to baselines. Here, UTRL [†] denotes the variant of UTRL using 4.5K programming tasks for training, for fair comparison to CURE.

    select 12 test cases from the ground-truth unit test for each task in the TACO dataset. As shown in Figure 11 (left), we format the target response by listing the 12 test cases without reasoning and use them directly as SFT labels.

- **SFT with $\mathcal{D}_{reason+UT}$**: As a training dataset, we use the 15,249 training task instances in TACO dataset, and we fine-tune Qwen3-4B. In order to label $\mathcal{D}_{reason+UT}$ with reasoning-annotated unit tests, we prompt Gemini-2.5-flash with the programming instruction to generate 12 test cases with reasoning. We then filter out functionally invalid test cases (i.e., test cases that fail under the ground-truth solution), and use the remaining valid test cases annotated with reasoning as SFT labels. We remark that this is replication of CodeRM (Ma et al., 2025), which utilized Llama3-70B-Instruct to label unit test with reasoning, and train the Llama3-8B-Instruct with the labeled dataset.

- **CURE**: For evaluating CURE and compare it with UTRL, we use the open-source ReasonFlux-7B-Coder, which is Qwen2.5-7B-Instruct fine-tuned with the CURE algorithm using 4.5K programming tasks from a subset of the CodeContest dataset (Li et al., 2022). In order to ensure fair comparison, we train Qwen2.5-7B-Instruct with UTRL using the same 4.5K training tasks (denoted as UTRL [†]), and compare the resulting model with ReasonFlux-7B-Coder. Additionally, following the evaluation setup of CURE (Wang et al., 2025a), for each programming instruction in evaluation set, we sample 16 test cases to form a single unit test, using the prompt format provided in the CURE paper.

## A.4 EVALUATION DETAILS

**Evaluation tasks**    Across all experiments, we use 945 competitive programming tasks in the evaluation split of the TACO dataset. The evaluation tasks span diverse difficulty levels, and each task is annotated with highly comprehensive unit test (i.e., ground-truth unit test) which comprises an average of 202.3 test cases. The tasks were collected from Codeforces, CodeChef, HackerRank, and HackerEarth, which are widely used competitive programming platforms, where the task distribution is described in Table 10.

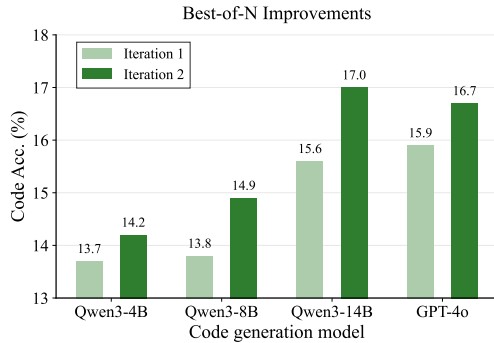

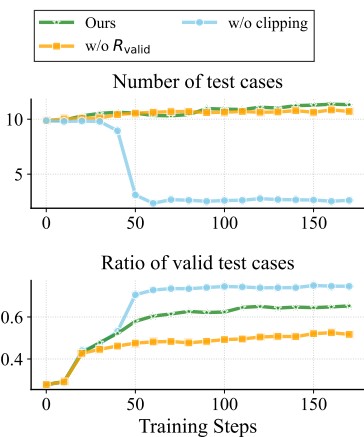

Figure 9: Best-of-N performance gains obtained by UTRL at the same iterations. Across 4 code generation models, iteration 2 results in improved best-of-N improvement compared to iteration 1.

Figure 10: Averaged number of test cases in the generated unit test and ratio of valid test cases across RL training steps. Ablated variants of UTRL results in generation of unit test with invalid test cases, or only few trivial test cases.

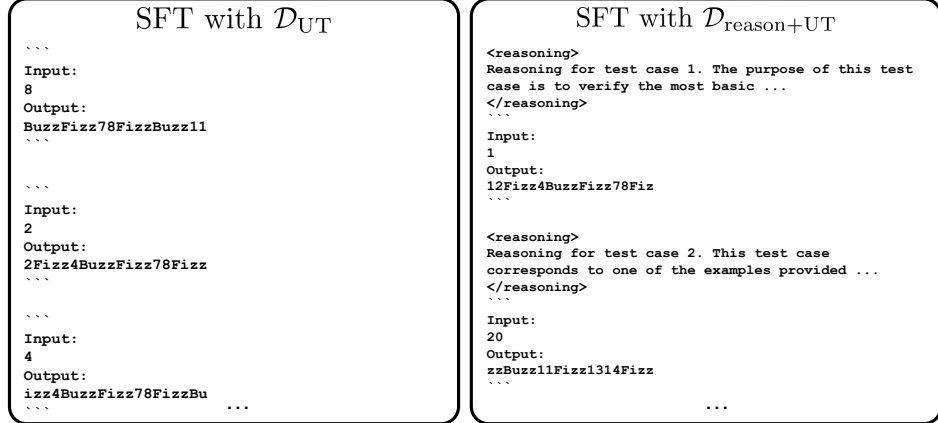

Figure 11: Examples of unit test annotations for two datasets for SFT baselines (SFT with $\mathcal{D}_{\mathrm{UT}}$ and SFT with $\mathcal{D}_{\mathrm{reason+UT}}$). $\mathcal{D}_{\mathrm{reason+UT}}$ is annotated by ground-truth unit tests, while $\mathcal{D}_{\mathrm{reason+UT}}$ is annotated by the teacher model (Gemini-2.5-flash).

**Best-of-N improvement**    For measuring Best-of-N improvement, we utilize LLM for sampling 32 code solutions, and select the best code solution by using the generated unit test as a code evaluator (i.e., select the code solution that passes the largest number of test cases in the generated unit test). We then evaluate code score and code accuracy regarding the selected code solution, using ground-truth unit tests provided in the TACO evaluation dataset. An intuition behind this evaluation is that a more discriminative unit test will identify and select higher-quality code among multiple candidate code solutions. In TACO evaluation set, we evaluate the Best-of-$N$ improvement on multiple code generator LLMs, Qwen3-4B, 8B, 14B, and GPT-4o. Additionally, in LiveCodeBench-v2 evaluation set, we evaluate the Best-of-$N$ improvement on Qwen3-4B and Qwen3-8B code generator LLMs.

**Unit test fidelity**    Unit test fidelity measures whether the generated unit tests induce code evaluation consistent with the code evaluation by the ground-truth unit tests. For each programming task, we first sample 128 code solutions using Qwen3-4B, 8B, 14B and GPT-4o (i.e., sample 32 code solutions using each LLM). We then evaluate code score of the 128 code solutions twice, once by using the ground-truth unit test, and once by using the generated unit test, which induces two score vectors length of 128 (i.e., number of code solutions under evaluation) and each score ranges from 0

| Methods | Optimizer | LR | Batch size | warmup | kl coef | sampling temperature |
|---|---|---|---|---|---|---|
| UTRL (iter 1) | AdamW | 1e-6 | 128 | 1e-2 | 1e-3 | 1.0 |
| UTRL (iter 2) | AdamW | 5e-6 | 128 | 1e-2 | 1e-3 | 1.0 |
| CURE | AdamW | 1e-6 | 128 | 1e-2 | 1e-2 | 1.0 |
| SFT w/ $\mathcal{D}_{\text{UT}}$ | AdamW | 1e-5 | 128 | 1e-2 | - | - |
| SFT w/ $\mathcal{D}_{\text{reason+UT}}$ | AdamW | 1e-5 | 128 | 1e-2 | - | - |

Table 9: Training hyperparameters of UTRL and baselines.

| | CodeForces | CodeChef | HackerRank | HackerEarth |
|---|---|---|---|---|
| # of tasks | 540 | 245 | 20 | 40 |

Table 10: Distribution of TACO evaluation tasks over multiple competitive programming platforms.

to 1. Finally, we compute Spearman's correlation between these two vectors to quantify how closely the generated unit tests reproduce the code evaluations of the ground-truth unit test.

### A.5 PROMPTS

In this section, we provide the prompts used in our experiments: (1) unit test generation, (2) code generation, and (3) rationalization, which were used for the RT baseline in Section 4.3.

---

**Unit test generation prompt**

**System prompt:**
```
You are an expert Python programmer capable of generating test cases
for Python programming tasks.
Given a programming task, generate several independent test cases with
corresponding reasoning.
Each test case should be independent of the others and sharply target
distinct corner cases so that arbitrary faulty code
solutions can be detected.
Before writing each test case, think deeply
about the input arguments that expose extreme or subtle edge cases,
and reason about the expected output.
After completing the reasoning process, generate the test case
in stdio format.
Specifically, your output should follow the format below:

<reasoning>
Reasoning for test case 1
First, reason about the input arguments that can discriminate an
incorrect code solution (e.g., edge cases).
Ensure the input arguments test aspects independent of
previous test cases.
Then, derive the expected output from the problem description.
</reasoning>
```
Input:
stdio format input 1

Output:
stdio format output 1
```

<reasoning>
Reasoning for test case 2
First, reason about the input arguments that can discriminate an
incorrect code solution (e.g., edge cases).
Ensure the input arguments test aspects independent of
previous test cases.
```

---

```
Then, derive the expected output from the problem description.
</reasoning>
```
Input:
stdio format input 2

Output:
stdio format output 2
```

...

<reasoning>
Reasoning for test case 12
First, reason about the input arguments that can discriminate an
incorrect code solution (e.g., edge cases).
Ensure the input arguments test aspects independent of
previous test cases.
Then, derive the expected output from the problem description.
</reasoning>
```
Input:
stdio format input 12

Output:
stdio format output 12
```

Ensure the following:
1. Do not include solution code in your response; generate
exactly 12 test cases.
2. For each test case, provide a detailed rationale.
3. Each test case must be independent; avoid duplicates.

**User prompt:**
Here is the problem description:

{problem_query}

Based on comprehensive reasoning, generate comprehensive unit test
involving several test cases for the given problem.
The test cases should cover various edge cases,
corner cases, and normal cases, at the same time, functionally correct.
```

---

### Code generation prompt

**System prompt:**
You are an expert Python programmer.
Based on the problem description, solve the coding
problem efficiently.
Think step by step, then write a Python solution that
solves the problem.
Follow the format below:

<reasoning>
Write your reasoning here.
</reasoning>

```python
Write your Python solution here. It should run with stdio-format input.
```
**User prompt:**
Here is the problem description:

{problem_query}

### A.6 LIMITATIONS & FUTURE DIRECTIONS

We outline the limitations of UTRL and promising directions for future research.

**Performance gap with ground-truth unit tests**   Although we show that UTRL is a promising direction for improving unit test generation capabilities of LLMs, a performance gap remains between UTRL-generated unit tests and ground-truth unit tests. As UTRL can be combined with arbitrary online RL algorithms, we believe that investigation on RL algorithm enabling better exploration, and combination of UTRL with the better RL algorithm will further improve the unit test generation performance.

**Extension to broad software engineering domains**   While our work conducts experiments on the competitive programming task domain, UTRL can be applied to broader software / programming domains. Scaling UTRL with large-scale instruction–code datasets covering a wider range of programming scenarios might be a promising direction for future work.

**Considerations for variable length unit tests**   UTRL trains the LLM to generate fixed number of test cases per unit test. Future work could explore framework that enables adaptive generation of variable-length unit tests depending on the complexity of the given programming task.

