# OpenReview forum: "Learning to Generate Unit Test via Adversarial Reinforcement Learning"
_ICLR.cc/2026/Conference — ICLR 2026 Poster_

### Official Review · Reviewer_NJwC · 2025-10-24

**Soundness:** 3
**Presentation:** 4
**Contribution:** 3
**Rating:** 6
**Confidence:** 3

**Summary:**

The authors proposed using reinforcement learning to jointly train a unit test generator and a code generator. The unit test generator is optimized for generating test cases that can discriminate gold code and model-generated code, while the code generator is optimized for pass rate. This eliminates the need for collecting expensive gold unit test and experimental results show that the proposed method has superior performance than SFT baselines and prompting frontier models to generate test cases.

**Strengths:**

1. The idea of modeling code generation and unit test generation as an adversarial setup is sound.
2. This setup removes the need for gold unit test data, which is particularly hard to collect.
3. Qwen3-4B trained with UTRL outperforms GPT4.1 and GPT4o in terms of unit test generation, which is a strong result.
4. The paper is well presented and is easy to follow.

**Weaknesses:**

1. The experiments are mainly conducted with Qwen3-4B, lacking analysis on the effectiveness across different models such as Llama, Gemma, or more code-specific ones like Qwen Code. I also wonder if the trained unit test generator is able to improve frontier models, when they are served as the code generator.
2. There is a hyperparameter, lambda, that balances the discrimination reward and the validity reward, which might need tuning. However, the analysis of this hyperparameter is missing.
3. There is little discussion on the iterative training part, where the authors only mention that the discrimination reward saturates at iter 1 and keeps improving at iter 2. But how many iterations are needed, and could we expect even more improvement after iteration 2 is not mentioned.

**Questions:**

1. In figure 3, the caption says “Second, the discrimination reward is defined as a ratio of sampled code solutions that do not pass at least one valid test case. In this figure, among 6 sampled code solutions, 4 code solutions (C1, C3, C4, C5) do not pass at least one valid test case, resulting in the discrimination reward of 0.667”, but the from the figure it seems like C1, C3, C4, C5 pass at least one valid test case. Do the authors mean “solutions that have at least one failure case”?
2. I wonder what the performance would be if we train the unit test generator without the adversarial setup, i.e., only train it using RL and the gold code solution to generate valid unit tests, then use it at test time to measure how much it could improve code generation.
3. In section 3.1, the author mentioned that the two key steps are repeated iteratively. How many iterations does it need to converge, and how is the performance changing over iterations?

---

> ### Author Response · Authors · 2025-11-21
>
> Dear Reviewer NJwC,
> We sincerely appreciate your efforts and thoughtful comments to help improve our manuscript. Below, we provide detailed responses to each of your comments.
>
> ---
>
> **[W1] Effectiveness of UTRL on different base LLMs**
>
>
>
> To address this concern, we trained Llama3.1-8B-Instruct with UTRL for 50 steps and evaluated its best-of-N performance. As shown in the table below, UTRL consistently improves unit test generation capability across different model architectures.
>
> |                                     | CodeLLM: Qwen3-4B |         |       CodeLLM: Qwen3-8B  |                 |
> |-------------------------------|-------------------|-----------------|-------------------------|-------------------------|
> |                                     | CodeAcc (%) | CodeScore | CodeAcc (%)        | CodeScore           |
> | Llama-3.1-8B-Instruct  |  7.8%           |    0.368       |     7.7%               |        0.38             |
> | Llama-3.1-8B-Instruct + UTRL  |      11.6%          |     0.475         |       11.9%     |   0.494     |
>
> Moreover, regarding the effectiveness of UTRL for improving frontier model performance, we clarify that unit tests generated via UTRL improve the best-of-N code generation performance of GPT-4o, as shown in Table 3 of Appendix A.2.
>
> ---
>
> **[W2, Q2] Ablation on lambda hyperparameter balancing discrimination reward and validity reward**
>
> In order to address the reviewer’s concern, we sweep over the hyperparameter $\lambda$, which controls the weight of the discrimination reward term, and analyze best-of-N improvement and ratio of valid test cases (i.e., validity ratio). In this experiment, we trained Qwen3-4B for 50 steps via UTRL with different choice of $\lambda$.
>
> As shown in the tables below, setting $\lambda = 0.0$ leads to improved validity ratio, while resulting in lower Best-of-N improvement, since the model is guided to generate test cases that are merely valid but indifferent to discriminativeness. Conversely, setting $\lambda = 1.0$ results in a substantial fraction of invalid test cases, which also degrades unit test quality measured by Best-of-N improvement. Setting $\lambda=0.85$ achieves the best unit test quality, balancing validity and discriminativeness of unit tests.  These results imply that both discrimination reward and validity reward are essential, and should be balanced appropriately to produce high-quality unit tests.
>
> |                     |      validity ratio (%)  |
> |------------------|----------------------------|
> | $\lambda$ = 0     |          64.3            |
> | $\lambda$ = 0.85  |      57.2         |
> | $\lambda$ = 1.0    |         49.7       |
>
> |                      | CodeLLM: Qwen3-4B |                       |       CodeLLM: Qwen3-8B   |                |
> |------------------|-----------------------|--------------------------|-------------------------|---------------------------|
> |                     | CodeAcc (%)     | CodeScore            | CodeAcc (%)        | CodeScore            |
> | $\lambda$ = 0   |       12.4        |    0.509                      |        13.1                 |     0.524                 |
> | $\lambda$ = 0.85 |   **13.7**        |   **0.514**                  |      **13.8**      |      **0.534**               |
> | $\lambda$ = 1.0  |   12.3        |   0.485                         |      11.5                  |      0.492               |
>
> ---
>
> **[W3, Q3] Discussion on iterative training**
>
> Regarding your question about how many iterations are needed, we first clarify that while single iteration (i.e., unit test generator LLM is trained to discriminate code generated by base code generator) was sufficient to yield a substantial performance gain as we demonstrated in Figure 9 of Appendix A.2, extending training to iteration 2 enabled performance that surpasses GPT-4.1.
>
> Additionally, to address your inquiry about further iterations, we extend the training to a third iteration. However, we did not observe a notable improvement in discrimination reward beyond the second iteration.
>
>
> ---
>
> **[Q1] Clarification on caption of Figure 3**
>
> We would like to clarify that the discrimination reward is computed based on the number of “code solutions **do not pass** at least one valid test case”, rather than “code solutions **pass** at least one valid test case”. For clarity, we’ve revised the caption in Figure 3 of the updated manuscript.

---

> > ### Comment · Reviewer_NJwC · 2025-11-25
> >
> > Thank the authors for the detailed responses. My comments have been addressed, and I don't have other major concerns. After checking the revised PDF and other reviews, I decided to raise my score to 8 and look forward to seeing this paper published at ICLR.

---

> > > ### Author Response · Authors · 2025-11-26
> > >
> > > Thank you very much for your feedback and valuable suggestions. We are happy to see that our response addressed your concern. Please feel free to reach out with any additional questions or suggestions.
> > >
> > > Many thanks,
> > > Authors

---

### Official Review · Reviewer_eXix · 2025-10-26

**Soundness:** 3
**Presentation:** 3
**Contribution:** 2
**Rating:** 6
**Confidence:** 4

**Summary:**

This paper introduces an RL method for training model to generate unit tests. Their method trains UT generator and code generator together in an adversarial loop: the UT generator is supposed to find faults, while the code generator is supposed to improve to pass more unit tests.

The method uses iterative RL training of 2 LLMs, with each model training off each other's signal, with the main contribution being a reward for UT generation, which informs the reward for code generation. The models are trained on examples from the TACO dataset.

The method is evaluated on TACO, with results showing improved UG generation as compared to base model, and as compared to an SFT model. The method is compared to multiple baselines:
- untrained models (closed and open-source)
- SFT model, trained on data filtered from Gemini 2.5
- compare to CURE UTs generated by a Qwen 2.5 7B model finetuned by CURE authors
- Ground-truth unit tests (ceiling)

Methods are evaluated according to best-of-N reranking with different models as generators and UT fidelity, which is the Spearman correlations between the ranking of several solutions according to a generated and ground truth unit tests.
The results indicate that UTs generated by the proposed method improve both Qwen-3-4B and Qwen-3-14B in best-of-32 and achieve higher fidelity. They also indicate that in an apples-to-apples comparison on Qwen-2.5-Instruct, the method outperforms CURE.

**Strengths:**

- Paper is clearly written and well-organized, and explanation of results is clear
- The method appears to work well as a verifier, with improvements even over proprietary models. The adversarial nature of the method is well-motivated.
- Unit test fidelity as a metric seems to be a novel contribution.
- Compares to recent baseline (CURE)
- Problem importance: unit test generation is an important problem

**Weaknesses:**

- no pass@1 results. From the way that the adversarial setup is described in the intro, I would have expected to see improvements to pass@1 for the adversarial generator. It seems like the evaluation (reranking@32) is geared towards the discriminator and test-time re-ranking, but does the pass@1 of the generator also improve via training? This is shown a bit in Fig 7 but should be expanded (since code score also is not quite the same thing as pass@1). Effectively, section 4.4 could be scaled up.
- OOD generalization: both train/test are TACO, so it's not clear that the method will generalize to other datasets. This might be something that is covered in the CodeContest experiment with Qwen-2.5 compared to CURE, but this should be made more formal/explicit. Moreover, it's not clear why the paper doesn't evaluate on the same data as CURE (which they use as a baseline) or UTGEN (which they cite).
- missing related work: https://arxiv.org/abs/2502.14948 also iteratively improves LLMs for solving and testing using RL with DPO.
- for CURE comparison: it would be nice to also see CURE applied to Qwen-3-4B
- limited model size: does this still result in improvements with stronger/larger models? It would also be nice to see a larger Qwen-3 model as a baseline for UT generation in Table 1 (since we don't know what size 4o and 4.1 are.)

**Questions:**

- L156-157: under (1), your goal is to discriminate sampled code from gold code. Doesn't this assume that the sampled code is incorrect? If so, does the current method still work when the generator is a stronger coding model?
- Qwen-3-4B: is this the instruct or base model? if it's the base model, do you still get improvements over instruct model?
- I would suggest adding experiments showing
	- more on pass@1
	- cross-dataset transfer
	- comparison to larger models
- a bit of a nit: since CURE came out in June, it seems odd to call it "concurrent work" (L097) and this doesn't seem necessary since the authors compare against it.

---

> ### Author Response · Authors · 2025-11-21
>
> Dear Reviewer eXix,
> We sincerely appreciate your efforts and thoughtful comments to help improve our manuscript. Below, we provide detailed responses to each of your comments.
>
> ---
>
> **[W1, Q3] Pass@1 accuracy of code generator**
>
> We first clarify that Code Acc (%) in Figure 7 refers to pass@1 accuracy, which measures whether the single solution generated by the code generator passes entire ground-truth test cases. As shown in figure 7 of our manuscript, pass@1 accuracy of Qwen3-4B code generator improves from 8.0% to 15.3% through 330 steps of UTRL training.
>
> Moreover, to further address your concern about lack of code generation baselines, we conduct comparison with following code generation baselines:
>
> * SFT with ground-truth code solution
> * RL using unit tests generated by GPT-4o
>
> As shown in the table below, the code generator trained via UTRL outperforms both baselines in terms of pass@1 accuracy. Specifically, as presented in Figure 7 of the updated manuscript, the pass@1 code accuracy of RL with GPT-4o-generated unit tests saturates around 11~13%, whereas UTRL continues to improve and reaches 15.3%, demonstrating the effectiveness of UTRL in training the code generator.
>
> |                      |  Pass@1 Acc (%)     |
> |------------------|-----------------------|
> | Qwen3-4B   |          8.0                |
> | Qwen3-4B + SFT       |        3.6           |
> | Qwen3-4B RL with GPT-4o unit tests   |  12.7       |
> | Qwen3-4B RL with UTRL unit tests       |  15.3   |
> | Qwen3-4B RL with GT unit tests           | 15.9   |
>
> ---
>
> **[W2, Q3] Evaluation on out-of-distribution benchmark**
>
> Thank you for the constructive suggestion. Since TACO evaluation set already includes majority of tasks in CodeContests and CodeForces test suite, we additionally evaluate UTRL on LiveCodeBench. As shown in the table below, Qwen3-4B trained via UTRL outperforms GPT-4.1 in best-of-N improvement, though with a slight decrease in unit test fidelity. Furthermore, Qwen2.5-7B trained via UTRL outperforms the same model trained via CURE, showing a trend consistent with results observed on the TACO evaluation suite.
>
> |                     | CodeLLM: Qwen3-4B |                        |       CodeLLM: Qwen3-8B |                             |
> |------------------|-----------------------|--------------------------|-------------------------|---------------------------|
> |                     | CodeAcc (%)     | CodeScore            | CodeAcc (%)        | CodeScore            |
> | Qwen3-4B   |       55.2%        |    0.704                  |        55.4                 |     0.714                   |
> | GPT-4.1      |   58.3%            |    0.732                   |        58.3                 |     0.735                  |
> | Qwen3-4B + UTRL  |   59.9%        |   0.744              |      59.3                   |      0.752               |
> | Qwen2.5-7B                |  44.6%           |    0.607       |     44.0%               |        0.617             |
> | Qwen2.5-7B + CURE  |  48.2%           |    0.660       |     47.7%               |        0.671             |
> | Qwen2.5-7B + UTRL   |  54.4%           |    0.686       |     52.4%               |       0.693              |
>
>
> |                          |      Unit test fidelity          |
> |------------------|-----------------------|
> | Qwen3-4B   |       0.650        |
> | GPT-4.1      |    0.771            |
> | Qwen3-4B + UTRL  |   0.752        |
> | Qwen2.5-7B                |  0.435           |
> | Qwen2.5-7B + CURE  |    0.560       |
> | Qwen2.5-7B + UTRL   |  0.589          |
>
> ---
>
> **[Q2] Clarification on base LLM**
>
> We clarify that Qwen3-4B, Qwen3-8B, and Qwen3-14B used in our experiments are all instruction-finetuned models.For clarity, we have detailed this point in Section 4.1 of the updated manuscript.
>
> ---

---

> ### Author Response · Authors · 2025-11-21
>
> **[W5, Q3] Improvements compared to larger models**
>
> To address your concern, we additionally evaluated the instruction-finetuned Qwen3-32B model as a unit test generator baseline. As shown in the table below, Qwen3-4B trained via UTRL clearly outperforms Qwen3-32B in Best-of-N improvement on the TACO evaluation suite. We have included this result in Table 1 of the revised manuscript.
>
> |                     | CodeLLM: Qwen3-4B |                        |       CodeLLM: Qwen3-8B  |                                     |
> |------------------|-----------------------|--------------------------|-------------------------|---------------------------|
> |                     | CodeAcc (%)     | CodeScore            | CodeAcc (%)        | CodeScore            |
> | Qwen3-32B  |   10.5            |             0.474          |        11.7                 |       0.484                |
> | Qwen3-4B + UTRL  |   14.2        |        0.536         |      14.9                   |      0.558               |
>
> Moreover, while evaluating Qwen3-32B trained via UTRL would indeed strengthen our results, the computational cost for training such a large model is beyond the scope of therebuttal period. Nevertheless, since Qwen3-4B trained via UTRL already outperforms the Qwen3-32B baseline, we expect that applying UTRL for training Qwen3-32B would yield even higher performance.
>
> ---
>
> **[W3] Missing related works**
>
> Thank you for pointing out the missing related work. We’ve updated the related works section in the updated manuscript.
>
> ---
>
> **[W4] CURE on Qwen3-4B**
>
> Thank you for the helpful suggestion. We agree that verifying whether UTRL outperforms CURE under Qwen3-4B would strengthen our work. However, due to limited time and substantial computational cost for RL, conducting such experiment was not feasible within the rebuttal period. Importantly, UTRL's advantage over CURE has been consistently demonstrated on Qwen2.5-7B-Instruct across both TACO and LiveCodeBench, providing strong evidence of its effectiveness.
>
> ---
>
> **[Q1] Assumption on correctness of the sampled code**
>
> We acknowledge that UTRL assumes the sampled code to be incorrect during training. Nevertheless, even when the code generator is comparatively strong, the unit test generator continues to receive meaningful learning signals unless the code generator always produces correct solutions for entire training tasks. In such cases, the unit test generator is still trained to discriminate at least one incorrect code, allowing the adversarial learning process to proceed effectively.

---

> > ### Comment · Reviewer_eXix · 2025-11-24
> > **Response to rebuttal**
> >
> > Thanks for these additional results and clarifications, which address some of my concerns especially about comparisons to past work.
> >
> > For the larger model results, I had hoped to see:
> > - the larger model being trained with UTRL
> > - the larger model being used as the code LLM
> >
> > While I understand that the first may be out of scope because of computational/time constraints, I think the second experiment would be a valuable addition (it would be impressive if the 4B UT generator could help improve a 32B code generator)

---

> > > ### Author Response · Authors · 2025-11-26
> > >
> > > Thank you for the valuable suggestion. Following your comment, we evaluate whether the Qwen3-4B unit test generator trained via UTRL can further improve the best-of-N performance of a larger code generator, Qwen3-32B. As shown in the table below, using unit tests generated by UTRL-trained Qwen3-4B significantly boosts the best-of-N sampling performance of Qwen3-32B code generator. This result demonstrates that the 4B unit test generator is capable of producing high-quality, discriminative unit tests that effectively evaluate and distinguish code generated by a substantially larger Qwen3-32B.
> > >
> > > | Unit test generated by: | CodeLLM: Qwen3-32B |                        |
> > >  |--------------------------------|--------------------------------|--------------------|
> > > |                                      | CodeAcc (%)                | CodeScore     |
> > > | Qwen3-32B                  |    13.3                            |      0.507         |
> > > | Qwen3-4B + UTRL       |   17.9                            |        0.592      |
> > >
> > > Importantly, as shown in the table below, we note that the pass@1 accuracy achieved by the Qwen3-4B code generator trained via UTRL already surpasses that of the Qwen3-32B baseline. It suggests that applying UTRL to larger code LLM has strong potential for further gains.
> > >
> > > |                      |  Pass@1 code acc (%)     |
> > > |------------------|-----------------------|
> > > | Qwen3-32B   |          9.5               |
> > > | Qwen3-4B trained via UTRL     |  15.3   |

---

> > > > ### Comment · Reviewer_eXix · 2025-11-26
> > > > **Updated response**
> > > >
> > > > Thanks for this updated result -- I think this is a strong result that is worth expanding and including in the paper. I have increased my score accordingly and recommend the paper be accepted.

---

> > > > > ### Author Response · Authors · 2025-12-01
> > > > >
> > > > > Thank you very much for your valuable feedbacks and comments. We are glad to see that our response addressed your concern. We will incorporate this updated result in our manuscript.
> > > > >
> > > > > Many thanks, Authors

---

### Official Review · Reviewer_qbAT · 2025-10-27

**Soundness:** 2
**Presentation:** 3
**Contribution:** 2
**Rating:** 4
**Confidence:** 4

**Summary:**

This paper presents UTRL, an adversarial RL loop training the language model to be code generator and a unit test generator. These components are trained in an adversarial manner with the UT generator acting as the discriminator -- distinguishing ground truth code solution from incorrect codes generated by the code generator while generating valid unit tests. The code generator on the other hand is optimized to have a higher pass rate over the generated unit tests. The paper makes use of the TACO dataset for code generation and verifies that their system leads to better quality unit tests and higher code generation performance when compared to different SFT and RL baselines (like CURE).

**Strengths:**

- The adversarial setting in the method intuitive and appears to be effective compared to the baselines (although the latter can be improved, see below)
- The paper is well-written, well-structured, and easy to follow. The problem or topic is well-motivated, and joint improvements in unit test generation and code generation could be of interest to the community.

**Weaknesses:**

- Missing references to related work, I think the scope of related works discussed is very myopic, and I would broadly discuss the following line of work:
    -  Self-play setups for LLMs: https://arxiv.org/abs/2407.19594, https://arxiv.org/abs/2401.01335, https://arxiv.org/abs/2502.14948
    - In my opinion, it is worth emphasizing the adversarial training setup in general which this method builds on, e.g. GANs (https://arxiv.org/abs/1406.2661), https://arxiv.org/abs/1607.02533, etc.

- Strength of SFT baseline:
    - In table 1 and 2, why not present the UT + Reason baseline? No discussion why the code-generation performance of UT + Reason baseline is lesser in Fig 5 (a) despite better code fidelity.
    - Alternate design, Prasad et al show that SFT supervision by training unit tests to be discriminative (they call attacking) and valid (with correct) outputs is effective. Sec 3.2 suggests such data is already collected during training, set of UTs which are both discriminative and valid. What happens when you train an SFT baseline with or without reasoning on this data? Would this remove the need for joint RL training?
- Lack of OOD generalization: The paper only tests on an in-domain test set of TACO dataset. Can the author show improvements in code-generation and UT fidelity transferring to CodeContests, LeetCode, MBPP, LiveCodeBench and other mainstream code-generation datasets.

**Questions:**

See above in addition to the questions below:
1. What is a tangible takeaway from Fig 7, how does the training trend contrast with other baselines?
2. Isn't the training online for the main results reported in Table 1, or is it iterative as well -- if so how is the number of iterations set? Can Fig 8 be extended to more iterations?
3. The paper repeatedly makes the claim that UTRL removes the need for ground-truth UT annotations. However, the training uses the regular code-generation recipe of instruction, gold solution. When these are available, ground-truth UTs can be readily collected by sampling multiple inputs from an LLM, rejecting invalid outputs. The authors should clarify what other kinds of annotations if any they are referring to.
4. As mentioned in Sec 3.1, you generate a set of UTs for each problem. Are the number of UTs roughly similar for various baselines?
5. Why not show the pass@1 accuracy of the code generator before and after training? Does that not improve and the improvement only come from best of n sampling? If not, could the performance be further improved by doing best-of-N sampling over the trained code-generator model or is it not effective due to the nature of adversarial training. An ablation separating the contribution of better code generation and UT generation would strengthen this work.

---

> ### Author Response · Authors · 2025-11-21
>
> Dear Reviewer qbAT,
> We sincerely appreciate your efforts and thoughtful comments to help improve our manuscript. Below, we provide detailed responses to each of your comments.
>
> ---
>
>
> **[W1] Limited coverage of related works**
>
> Thank you very much for pointing out the missing related works. Based on your suggestion, we revised the related works section in our updated manuscript.
>
> **[W2 - 1] Lack of discussion on SFT baselines**
>
> Regarding the questions about absence of SFT w/ $D_\text{reason+UT}$ baseline in Table 1, we would like to clarify that this setting was excluded from Table 1 since its training relies on unit tests generated by Gemini-2.5-flash, making it an unfair comparison with other methods. Therefore, results for this baseline are instead discussed in Figure 5. Nevertheless, for completeness, we additionally included a table presenting best-of-N improvement of SFT w/ $D_\text{reason+UT}$ in Appendix A.2.
>
> Moreover, to address your concern about the lack of comparative discussion regarding two SFT baselines (SFT w/ $D_\text{UT}$ and SFT w/ $D_\text{reason+UT}$), we provide additional clarification. We observe that the SFT w/ $D_\text{reason+UT}$ baseline tends to generate fewer highly discriminative test cases compared to SFT w/ $D_\text{UT}$, as it is trained on synthetic unit tests generated by Gemini-2.5-Flash rather than highly discriminative ground-truth unit tests. Since Best-of-N improvement is largely determined by the presence of highly discriminative test cases, SFT w/ $D_\text{reason+UT}$ exhibits lower Best-of-N improvement compared to SFT w/ $D_\text{UT}$. We’ve incorporated this discussion in Section 4.3 of our updated manuscript.
>
> ---
>
> **[W2 - 2] Comparison to alternative SFT approach**
>
> Thank you for suggesting an interesting approach to train a unit test generator.
> We believe your suggestion corresponds to a best-of-N distillation strategy, which is proven to be a simple yet effective technique for improving LLM performance across various domains [1, 2]. Following your suggestion, we constructed an SFT training dataset by sampling 8 unit tests from the unit test generator LLM per task instance, and selecting the one that maximizes a weighted sum of the discrimination reward and validity reward. We then fine-tuned the unit test generator using this curated SFT dataset.
>
> As shown in the table below, although the best-of-N distillation approach yields performance comparable to SFT w/ $D_\text{UT}$ baseline even without requiring ground-truth unit tests for training, UTRL achieves superior performance.
>
> Additionally, we would like to emphasize that this best-of-N distillation strategy still necessitates our discrimination reward for evaluating the unit tests, and the design of the discrimination reward is one of the core contributions of UTRL.
>
> |                     | CodeLLM: Qwen3-4B |                        |       CodeLLM: Qwen3-8B  |                 |
> |------------------|-----------------------|--------------------------|-------------------------|---------------------------|
> |                     | CodeAcc (%)     | CodeScore            | CodeAcc (%)        | CodeScore            |
> | Qwen3-4B + SFT w/ $D_\text{UT}$   |     11.7   |    0.457    |    11.7    |  0.458        |
> | Qwen3-4B + Best-of-N distillation |    10.2     |      0.423         |      9.9      |     0.437                |
> | Qwen3-4B + UTRL                 |     14.2    |     0.536       |   14.9      |     0.558                  |
>
> [1] Sessa et al., “BOND: Aligning LLMs with Best-of-N Distillation”, arXiv (2024)
> [2] Chow et al., “Inference-Aware Fine-Tuning for Best-of-N Sampling in Large Language Models”, arXiv (2024)
>
> ---

---

> ### Author Response · Authors · 2025-11-21
>
> **[W3] Evaluation on out-of-distribution benchmark**
>
> Thank you for this constructive suggestion. Since the TACO evaluation suite includes the majority of problems from CodeContests and LeetCode, we conducted additional evaluation on LiveCodeBench. As shown in the table below, Qwen3-4B trained via UTRL outperforms GPT-4.1 in best-of-N improvement, though with a slight decrease in unit test fidelity. Furthermore, Qwen2.5-7B trained via UTRL outperforms the same model trained via CURE, showing a trend consistent with results observed on the TACO evaluation suite.
>
> |                     | CodeLLM: Qwen3-4B |                        |       CodeLLM: Qwen3-8B  |                         |
> |------------------|-----------------------|--------------------------|-------------------------|---------------------------|
> |                     | CodeAcc (%)     | CodeScore            | CodeAcc (%)        | CodeScore            |
> | Qwen3-4B   |       55.2%        |    0.704                  |        55.4                 |     0.714                   |
> | GPT-4.1      |   58.3%            |    0.732                   |        58.3                 |     0.735                  |
> | Qwen3-4B + UTRL  |   59.9%        |   0.744              |      59.3                   |      0.752               |
> | Qwen2.5-7B                |  44.6%           |    0.607       |     44.0%               |        0.617             |
> | Qwen2.5-7B + CURE  |  48.2%           |    0.660       |     49.7%               |        0.671             |
> | Qwen2.5-7B + UTRL   |  54.4%           |    0.686       |     52.4%               |       0.693              |
>
>
> |                   |    Unit test fidelity            |
> |------------------|-----------------------|
> | Qwen3-4B   |       0.650        |
> | GPT-4.1      |    0.771            |
> | Qwen3-4B + UTRL  |   0.752        |
> | Qwen2.5-7B                |  0.435           |
> | Qwen2.5-7B + CURE  |    0.560       |
> | Qwen2.5-7B + UTRL   |  0.589          |
>
> ---
>
> **[Q5] Pass@1 accuracy for code generation**
>
> We would like to first clarify that Code Acc (%) in Figure 7 of our manuscript refers to pass@1 accuracy, which measures whether the single solution generated by the code generator passes entire ground-truth test cases. As shown in figure 7 of our updated manuscript, pass@1 accuracy of Qwen3-4B code generator improves from 8.0% to 15.3% after 330 steps of UTRL training. For clarity, we’ve revised Figure 7 and Section 4.4 of the updated manuscript.
>
> Moreover, we would like to remark that best-of-N improvement reported in Table 1 isolates the contribution of the improved unit test generator. Specifically, in the best-of-N improvement evaluation, the code generator LLM is kept fixed, and the unit tests produced by UTRL are used to select the best one among N solutions sampled by the fixed code generator. By measuring the accuracy and score of the selected code solution, we provide an isolated assessment over the unit tests produced by UTRL.
>
> ---
>
> **[Q1] Additional baseline for code generation**
>
> Regarding the question about the takeaway from Figure 7, the main point is that the code generator trained adversarially against the unit test generator achieves pass@1 code accuracy nearly matching the performance of a code generator trained with ground-truth unit tests via RL, which we consider as our upperbound.
>
> Moreover, to further address your concern about the lack of comparison with code generation baselines, we conduct a comparison with the following additional baselines:
>
> * SFT with ground-truth code
> * RL with unit tests generated by GPT-4o
>
> As shown in the table below, the code generator trained via UTRL outperforms both baselines in terms of pass@1 code accuracy. Moreover, as presented in Figure 7 of the revised manuscript, the pass@1 accuracy of RL with GPT-4o-generated unit tests saturates around 11~13%, whereas UTRL continues to improve and reaches 15.3%, demonstrating the effectiveness of UTRL in training the code generator.
>
> |                      |  Pass@1 Acc (%)   |
> |------------------|-----------------------|
> | Qwen3-4B   |          8.0               |
> | Qwen3-4B + SFT       |      3.6            |
> | Qwen3-4B RL with GPT-4o unit tests   |      12.7         |
> | Qwen3-4B RL with UTRL unit tests (ours)       |  15.3   |
> | Qwen3-4B RL with GT unit tests (upperbound)  | 15.9   |

---

> ### Author Response · Authors · 2025-11-21
>
> **[Q2] Discussion on additional iteration**
>
> Regarding the question about the main result in Table 1 and the number of iterations, we clarify that the Qwen3-4B was trained for two iterations. Moreover, to address your inquiry about the effect of additional training iterations, we extend the training to a third iteration. However, we did not observe a notable improvement in the discrimination reward beyond the second iteration.
>
> ---
>
> **[Q3] Why the annotation of high-quality unit test is costly**
>
> We agree that, given ground-truth code, it is possible to collect a set of valid test cases. However, functional validity of test cases does not guarantee that the test cases constitute optimal unit tests. As we discussed in Section 3.1 of our manuscript, effective unit tests must not only be functionally valid but also sufficiently discriminative to detect subtle faults in imperfect code solutions, which makes constructing such test cases challenging.
>
> ---
>
> **[Q4] Number of test cases in generated unit test**
>
> As elaborated in Appendix A.3 and Appendix A.5, we set the number of the test cases in the generated unit tests as 12 for all methods. As an exception, for CURE, we sampled 16 test cases per task following its evaluation protocol.

---

> > ### Author Response · Authors · 2025-11-26
> >
> > Dear Reviewer qbAT,
> >
> > Thank you again for your time and efforts in reviewing our paper.
> >
> > As the discussion period draws close, we kindly remind you that seven days remain for further comments or questions. We would appreciate the opportunity to address any additional concerns you may have before the discussion phase ends.
> >
> > Thank you very much!
> >
> > Many thanks, Authors

---

### Official Review · Reviewer_R3AP · 2025-10-31

**Soundness:** 3
**Presentation:** 3
**Contribution:** 2
**Rating:** 2
**Confidence:** 4

**Summary:**

This paper introduces UTRL, an adversarial reinforcement learning framework for training a Large Language Model (LLM) to generate high-quality unit tests. The framework co-trains two models: a unit test generator ($M_{\text{UT}}$) and a code generator ($M_{\text{code}}$). The core idea is that $M_{\text{UT}}$ is rewarded for creating tests that can distinguish between code generated by $M_{\text{code}}$ and a ground-truth solution ($C^*$), while $M_{\text{code}}$ is rewarded for generating code that passes the tests from $M_{\text{UT}}$. The main contribution is a method to train a unit test generator without requiring ground-truth unit tests, relying only on instruction-code pairs. Experiments on the TACO dataset show that the resulting unit test generator outperforms supervised fine-tuning (SFT) and proprietary models in producing high-quality tests.

**Strengths:**

1. The experimental setup is rigorous. The authors compare their method against a good set of baselines, including various LLMs, SFT, and the recent CURE framework. The introduction of "best-of-N improvement" and "unit test fidelity" as evaluation metrics is well-reasoned and provides a quantitative assessment of the quality of the generated tests.
2. The paper is clearly written and well-structured. The UTRL framework is explained effectively with figures, an algorithm description, and formal reward definitions, making the methodology easy to understand.
3.  The work addresses the important and challenging problem of automated unit test generation. Improving the quality of generated tests could provide a more reliable and scalable feedback mechanism for training and verifying code-generating LLMs.

**Weaknesses:**

The central claims of the paper rest on a fundamental assumption that severely limits its practical applicability and scalability.

1.  Unrealistic Requirement of Ground-Truth Code: The entire training process is critically dependent on having access to a ground-truth, executable correct solution ($C^*$) for every training instance. This assumption is unrealistic for the vast majority of programming tasks. The primary value of automated code generation lies in solving problems for which a solution is not already known. If a correct, executable `C*` is available, the need to train a new code generator is greatly diminished. This requirement makes it impossible to scale the method using large, readily available datasets of code from sources like GitHub, confining it to niche, curated datasets.

2. No Improvement in Downstream Code Generation: The ultimate goal of generating better unit tests is often to improve the performance of code generators. However, the paper's own results in Figure 7 demonstrate that this goal is not achieved. The code generator trained adversarially with UTRL's generated tests performs comparably to, but slightly *worse than*, a model trained with rewards from the ground-truth unit tests. This suggests that the complex adversarial framework is, at best, a proxy for an oracle that it cannot outperform. While it successfully creates a high-quality unit test generator, it fails to show that these generated tests provide a superior training signal that pushes the code generator beyond existing capabilities. The main loop does not seem to create a net performance gain for the overall system.

3.  Overstated Claims on Scalability: Given the reliance on $C^*$ and the lack of improvement for the code generator, the claim that this method can enable better scaling of LLM training beyond current SOTA methods is not well-supported. The need for verified, executable solutions for every training problem is a far more significant bottleneck than the need for unit tests, fundamentally hindering scalability.

4. The proposed adversarial framework was proposed before in other similar scenario where an external interpreter is available (e.g. for Lean with STP [1])

[1] Dong, Kefan, and Tengyu Ma. "Stp: Self-play llm theorem provers with iterative conjecturing and proving." arXiv preprint arXiv:2502.00212 (2025).

**Questions:**

1.  The framework's dependency on a ground-truth executable solution appears to be its greatest limitation. Could the authors elaborate on the practical applicability of this method outside of curated competitive programming datasets where canonical solutions are available? How could this framework be adapted for a more realistic scenario where $C^*$ does not exist?

2.  Figure 7 indicates that the UTRL-trained code generator does not surpass the performance of an agent trained with ground-truth unit tests. Does this imply that the primary contribution is the creation of a standalone, high-quality unit test generator, rather than a superior end-to-end training pipeline for code generation? What is the practical advantage of the adversarial loop if the resulting code agent's performance does not improve beyond the oracle baseline?

---

> ### Author Response · Authors · 2025-11-21
>
> Dear Reviewer R3AP,
> We sincerely appreciate your efforts and thoughtful comments to help improve our manuscript. Below, we provide detailed responses to each of your comments.
>
> ---
>
> **[W1, W3, Q1] Clarification of problem setup and assumption on ground-truth code**
>
> We respectfully disagree that requiring ground-truth code for training is unrealistic. Large-scale instruction-code datasets [1, 2] are widely available and routinely used for post-training code LLMs, and prior works on training LLMs for unit test generation [3, 4] also assumes access to ground-truth code solutions.
>
> More importantly, while UTRL uses ground-truth code during training, the trained unit test generator is applied to unseen problems where no ground-truth solution is available. This mirrors how code generators are trained on existing solutions but evaluated on new problems. All our evaluations are conducted on unseen problems or even unseen domains to demonstrate generalization.
>
>
> [1] Lambert et al., “Tulu 3: Pushing frontiers in open language model post-training”, COLM 2025
>
> [2] Muennighoff et al., “OctoPack: Instruction Tuning Code Large Language Models”, NeurIPSw 2023
>
> [3] Prasad et al., “Learning to Generate Unit Tests for Automated Debugging”, COLM 2025
>
> [4] Ma et al., “Dynamic Scaling of Unit Tests for Code Reward Modeling”, ACL 2025
>
> ---
> **[W2, Q2] Comparison to code generator trained via oracle unit test**
>
> We clarify that, in our setting, a code generator trained with oracle (ground-truth) unit tests represents an upper bound on achievable performance. These oracle tests in Figure 7 are manually curated, comprehensive test suites that have undergone careful verification to ensure broad semantic and edge-case coverage. Such high-quality test suites are expensive to build and rarely available for real-world tasks. Given this context, the key contribution of UTRL is not to outperform the oracle baseline, but to match its performance while eliminating the need for manual test construction. Our results show that UTRL-generated unit tests guide code generator to achieve performance comparable to this oracle upper bound.
>
> Regarding the adversarial loop, while the UTRL-trained code generator does not surpass the oracle baseline, it shows clear improvement over the initial code generator. This improvement is crucial: as the code generator becomes stronger, the unit test generator must produce more comprehensive and challenging tests to maintain discrimination. This dynamic drives the unit test generator toward oracle-quality test generation without manual curation.
>
> ---
>
> **[W4] Existence of relevant method**
>
> Thank you for pointing out this relevant work. While STP and UTRL both employ self-play between two LLMs, our core contribution of discriminative reward design for unit test generation differs fundamentally from STP's prover-conjecturer framework in both objective and mechanism.

---

> > ### Author Response · Authors · 2025-11-26
> >
> > Dear Reviewer R3AP,
> >
> > Thank you again for your time and efforts in reviewing our paper.
> >
> > As the discussion period draws close, we kindly remind you that seven days remain for further comments or questions. We would appreciate the opportunity to address any additional concerns you may have before the discussion phase ends.
> >
> > Thank you very much!
> >
> > Many thanks, Authors

---

### Author Response · Authors · 2025-11-21
**General Response**

We deeply appreciate your time and effort in reviewing our manuscript. As highlighted by reviewers, UTRL is an effective adversarial reinforcement learning framework to train LLMs for unit test generation, which is a crucial and challenging problem (all reviewers).
Our paper clearly identifies challenges in collecting high-quality unit tests to train LLMs for unit test generation (NJwC), and proposes an intuitive RL-based adversarial training method that allows LLMs to produce discriminative unit tests even without access to high-quality test suites for training LLMs (qbAT, NJwC) with clear presentations (all reviewers). Under a rigorous experimental setup (R3AP, eXix) and carefully designed evaluation metrics (R3AP, eXix), UTRL achieves strong empirical performance in unit test generation even outperforming proprietary LLMs (eXix, NJwC), and also improves code generation performance.

We appreciate the reviewer’s insightful comments on our manuscript. In response to the questions and concerns the reviewers raised, we have carefully revised and enhanced the manuscript with the following additional experiments and discussions:

* Addressing the missing related works (Section 2)
* Clarification on open-source LLM baselines (Section 4.1)
* Comparison with Qwen3-32B (Section 4.2)
* Additional discussion on performance of SFT baselines (Section 4.3)
* Improving discussion and experiments regarding UTRL for code generation (Section 4.4)
    - Comparison with additional code generation baselines (SFT, RL with unit tests generated by frontier models)
    - Updating figure 7
    - Clarification on evaluation metric for code generation.
* Additional discussion on iterative training (Section 4.5)
* Comparison to best-of-N distillation method (Appendix A.2.1)
* Generalization experiment on out-of-distribution benchmark (Appendix A.2.2)
* Experiment under different LLM (Appendix A.2.3)
* Ablation study regarding hyperparameter balancing discrimination reward and validity reward (Appendix A.2.4)

These updates are temporarily highlighted in “red” for your convenience. We strongly believe that UTRL can be a useful addition to ICLR community, particularly because reviewers’ constructive comments enhanced the manuscript. Thank you very much,
Authors

---

### Meta-Review · Area_Chair_Sav3 · 2026-01-07

**Summary:**

This submission proposes an adversarial RL loop that co-trains a unit-test generator and a code generator. The paper received reviews from four reviewers, resulting in divergent scores. Most reviewers recognized that automatic unit test generation is crucial and challenging, the adversarial setup is technically sound, and the quality of writing and presentation. The core of the main concerns related to a lack of comprehensiveness of the experimental work, such as the comparisons with extra datasets, more baselines/scales of models, and an ablation study about the adversarial training.

The rebuttal addressed these concerns by presenting additional evaluation and explanations, enabling two reviewers to raise their scores after the discussion (eXix and NJwC, from 6 to 8). The other two reviewers (R3AP and qbAT), however, did not participate in the discussion, but their core concerns, such as the reliance on ground-truth code solution, and more baselines and OOD evaluation, were well addressed.

Therefore, AC is to recommend acceptance at this stage.

**Reviewer Concerns:**

Addressed:
1. Evaluation Scope: Added OOD evaluation on LiveCodeBench, showing consistent improvements.
2. Code Generation Performance: Clarified and expanded results to show that the UTRL-trained code generator.
3. Model Scale: Demonstrated that a small 4B unit-test generator trained with UTRL can improve the output of a much larger 32B code model.
4. Baselines: Added more baselines.
5. Ablation Study: Provided an ablation study.

Unresolved :
1. Full UTRL training on ≥14 B models.
2. Public code release date.

**Reviewer Scores:**

Reviewer R3AP could have changed its score from  2 to 4, since concerns regarding the reliance on ground-truth code solutions were partially addressed.

Review qbAT could have changed its score from  4 to 6, since the requests for more baselines and OOD evaluation were addressed.

Reviewer eXix and NJwC have indicated that their scores were raised.

---

### Decision · Program_Chairs · 2026-01-26

Accept (Poster)